# A systematic review: Role of dietary supplements on markers of exercise-associated gut damage and permeability

Sarah Chantler[1,2]*, Alex Griffiths[3], Jamie Matu[3], Glen Davison[4], Adrian Holliday[1,5], Ben Jones[1,6,7,8,9]

1 Carnegie Applied Rugby Research (CARR) Centre, Carnegie School of Sport, Leeds Beckett University, Leeds, United Kingdom, 2 Yorkshire Carnegie Rugby Union Club, Leeds, United Kingdom, 3 School of Clinical and Applied Sciences, Leeds Beckett University, Leeds, United Kingdom, 4 Endurance Research Group, School of Sport and Exercise Sciences, University of Kent, Canterbury, United Kingdom, 5 Human Nutrition Research Centre, Population Health Sciences Institute, Newcastle University, Newcastle upon Tyne, United Kingdom, 6 School of Science and Technology, University of New England, Armidale, NSW, Australia, 7 Division of Exercise Science and Sports Medicine, Department of Human Biology, Faculty of Health Sciences, the University of Cape Town and the Sports Science Institute of South Africa, Cape Town, South Africa, 8 Leeds Rhinos Rugby League Club, Leeds, United Kingdom, 9 England Performance Unit, Rugby Football League, Leeds, United Kingdom

* s.a.chantler@leedsbeckett.ac.uk

**Data Availability Statement:** All relevant data are within the paper and its Supporting Information files.

## Abstract

Nutrition strategies and supplements may have a role to play in diminishing exercise associated gastrointestinal cell damage and permeability. The aim of this systematic review was to determine the influence of dietary supplements on markers of exercise-induced gut endothelial cell damage and/or permeability. Five databases were searched through to February 2021. Studies were selected that evaluated indirect markers of gut endothelial cell damage and permeability in response to exercise with and without a specified supplement, including with and without water. Acute and chronic supplementation protocols were included. Twenty-seven studies were included. The studies investigated a wide range of supplements including bovine colostrum, glutamine, probiotics, supplemental carbohydrate and protein, nitrate or nitrate precursors and water across a variety of endurance exercise protocols. The majority of studies using bovine colostrum and glutamine demonstrated a reduction in selected markers of gut cell damage and permeability compared to placebo conditions. Carbohydrate intake before and during exercise and maintaining euhydration may partially mitigate gut damage and permeability but coincide with other performance nutrition strategies. Single strain probiotic strains showed some positive findings, but the results are likely strain, dosage and duration specific. Bovine colostrum, glutamine, carbohydrate supplementation and maintaining euhydration may reduce exercise-associated endothelial damage and improve gut permeability. In spite of a large heterogeneity across the selected studies, appropriate inclusion of different nutrition strategies could mitigate the initial phases of gastrointestinal cell disturbances in athletes associated with exercise. However, research is needed to clarify if this will contribute to improved athlete gastrointestinal and performance outcomes.

**Funding:** The author(s) received no specific funding for this work.

**Competing interests:** SC is employed by the Yorkshire Carnegie Rugby Union Club. BJ is employed by the Leeds Rhinos Rugby League Club and the Rugby Football League. GD has received unrelated research funding and honorarium from PanTheryx Inc., who supply bovine colostrum products to U.S., European, and Asia-Pacific markets, and received funding from Neovite Colostrum UK in 2008, 2010, and 2013. There are no patents, products in development or marketed products to declare. This does not alter our adherence to PLOS ONE policies on sharing data and materials.

## Introduction

Gastrointestinal disturbances are common amongst endurance athletes and can be experienced as mild discomfort or be race ending [1, 2]. Due to the sometimes unpredictable and deleterious effects of gastrointestinal discomfort on competition or training performance, there has been an effort to understand the aetiology. In this regard, exercise-associated gastrointestinal syndrome (EAGS), originally termed by Costa and colleagues [3], maps the proposed physiological and neuroendocrine response of the gastrointestinal system to acute exercise. Subsequently, evaluating nutritional strategies to attenuate gastrointestinal injury and its consequences in relevant populations has also received considerable attention in the literature.

The interaction between exercise, the gastrointestinal lining and the experience of gastrointestinal symptoms (GIS) is complex. Exercise-associated splanchnic hypoperfusion leads to localised hypoxia and damage of the gastric endothelial cells [4]. Endothelial cell damage increases permeability via disruption and dysregulation of tight junction proteins [5, 6]. Increased permeability has been associated with changes in urinary metabolites [7] and inflammatory markers [8]. Runners with exercise-associated collapse during a marathon had intestinal fatty acid binding protein (i-FABP) levels in excess of 15,000 ng/l compared to the mean of 2593 ng/ml in healthy marathon finishers [9]. As such, a recent meta-analysis showed endurance exercise has a significant and independent effect on indirect markers of gut damage and permeability [10]. These changes can result in a larger risk of bacterial translocation across the abnormally permeable gut lining and an increase in endotoxin levels [11]. This in turn may trigger an increased immune response, which has been found to be amplified in the heat [11, 12]. Endotoxin levels have been associated with GIS, such as nausea in ultra-endurance athletes [13], or long distance heated running [11] but not in shorter duration exercise ($<$150 minutes) [14]. In addition, exercise stimulates the sympathetic neuroendocrine pathways, which reduces gastric emptying and overall motility, possibly exacerbating any risk of gastric distress. Gastroparesis or 'slosh stomach' may contribute to reduced nutrient absorption or gastrointestinal cell injury, but the overall rates and consequences in athletes are unknown [15]. Research continues to build a comprehensive picture around the various aspects of the cascade and its possible mitigating factors.

As there are multiple mechanisms involved in EAGS, there is no single strategy to mitigate the impact of exercise. There has been a recent surge in studies examining the impact of various dietary supplements on markers of endothelial cell gut damage and permeability. Targeted dietary supplementation has the potential to mitigate the initial cell damage in response to exercise and hence arrest the preliminary stages of the exercise-associated gastrointestinal syndrome. In theory, any reductions in this initial phase may contribute to better gastrointestinal outcomes over the duration of an exercise bout. Proposed nutrition support, including supplementation could include nutrients that improve perfusion, increase antioxidant capacity of endothelial cells, improve tolerance to ischemia or heat, improve the localised immune response, or strengthen the structural integrity and energy availability of the endothelial cells and tight junctions. Further, the impact of chronic or acute supplementation protocols with similar supplements has not been considered.

Given the increase in research with a large variety of supplements used across a wide range of study designs, it is important to synthesise these data to provide clarity on the current literature and strengthen evidence-based practice. A recent review has considered specific nutrient-gut interactions around heated environments [12], but not at thermoneutral ranges. This study aimed to systematically review the literature examining the use of supplements, where gut damage and/or permeability have been assessed in response to exercise.

## Methods

This systematic review was prospectively registered with the PROSPERO database (CRD42020168256) and was completed in accordance with PRISMA (Preferred Reporting Items for Systematic Review and Meta-analysis) guidelines (PRISMA checklist included in S3 File) [16]. Ethics was approved by University Human Research Ethics committee at Leeds-Beckett University (Reference: 91711).

### Literature search

Pubmed and the Cochrane Library, as well as MEDLINE, CINAHL and SPORTDiscus via EBSCOhost were searched through to the 1<sup>st</sup> of February 2021. Keywords searches were performed for: 'gut', 'gastrointestinal', 'GI', 'intestines', 'intestinal', 'mucosal', 'splanchnic', 'permeability', 'leaky', 'hyperpermeability', 'function', 'dysfunction', 'injury', 'exercise', 'training', 'endurance', 'physical activity', 'microbiota', 'microbiome', gastrointestinal microbiome'. Reference lists of eligible studies and review articles were also searched. Publication date and language restrictions were not applied. Details of the specific search strategy for each database can be found in supplementary material (S1 File).

### Inclusion criteria

Studies were included if the participants were healthy, 18–65 years of age, without any history of gastrointestinal illness or any other inflammatory, metabolic, cardiovascular, neurological, or psychological disease(s). These criteria were selected to target participants free of any disease or age-related outcomes that may confound the response to exercise or to a dietary supplement. Studies were required to include a biochemical measure of gastrointestinal damage or permeability. Studies that only examined markers associated to supplementation (e.g. fecal zonulin collected prior to exercise trial) or generalised inflammatory markers were not included as these may not be considered to be in response to exercise or gut specific [17–19]. Specific markers that were considered included intestinal fatty acid binding protein and ingested saccharides (dual saccharide absorption test). I-FABP is present in gastrointestinal endothelial cells and is involved in the transport of fatty acids. An increase in plasma concentrations of i-FABP has been linked to gut ischemia and endothelial cell disruption [20]. Urinary or plasma saccharide absorption tests rely on the different molecular sizes of various non-digestible saccharides (lactulose, mannose, rhamnose) to illustrate disturbances in transcellular transport [21]. Lactulose (disaccharide) absorption and appearance in the urine/plasma will increase with disturbances in endothelial gap junctions while mannitol/rhamnose (monosaccharide) absorption is used as a normalising factor for lactulose since it utilises paracellular transport. An increase in the ratio of the larger disaccharide to the smaller monosaccharide illustrates and increase in paracellular passage and hence intestinal permeability. There are a range of other biomarkers assessing GI damage [22, 23], of which none have been used regularly outside of diseased populations. It is standard practice for participants to avoid alcohol, spicy food, non-steroidal anti-inflammatories, caffeine or high intensity exercise in the day preceding exercise trials [3, 21].

Based on the definition of a supplement by the International Olympic Committee (IOC) position stand, supplemental macronutrients given as sports foods or supplements in addition to the standard diet could be defined as a 'supplement'. Supplements, can include any food, or substance derived from food that is taken in addition to the existing diet [24]. Recent studies have examined the impact of weeklong exposure to gluten-containing/gluten-free diets [25], or high low fermentable oligo-, di-, monosaccharides and polyol (FODMAP) diets (6 and 7 days) on markers of gut damage [26, 27]. Since these are food-based approaches to change the

composition of the diet, rather than to supplement the existing diet, they were not included in the review. For further reading, Lis et al (2019) and Costa et al (2020) have reviewed current research on FODMAPs thus far [12, 28]. In addition, there is evidence that water intake during exercise may provide some protection against exercise induced endothelial damage compared to fluid restriction [29]. Therefore, as an additional consideration for this review, studies were included where water was the experimental condition (supplement) compared to fluid restriction. These data may highlight considerations when using water as a placebo in future research.

Both acute (pre-exercise or during exercise) and chronic (multiple days) supplementation protocols were included. In order to assess the impact of both exercise and supplementation, studies were required to have a treatment and a placebo condition with data collection pre- and post-exercise (i-FABP), or rest compared to exercise (dual saccharide absorption test (DSAT)). The studies that included DSAT in their methodology but had no resting control were excluded from data extraction for that marker [30–33]. Studies were excluded if the post-exercise measure for the urinary saccharides was not within 24 hours of exercise as this is unlikely to be an appropriate proxy for exercise-associated permeability [34, 35]. Studies were included with a variety of exercise protocols (single mode, multi-mode, with/without a time trial) to broaden the scope of the review. Training based studies that did not include an exercise trial were not included [36]. No restrictions were placed on the training status or sex of the participants as the limited studies available do not justify this [37, 38].

Two researchers (SC and AG) independently assessed studies for inclusion and later compared notes to reach a mutual consensus. Disagreements about the eligibility of any particular studies were resolved by a third reviewer (GD). Potential studies that could be included based on their title or abstract were retrieved in full-text and reviewed against the inclusion/exclusion criteria independently by two researchers (SC and AG) with a third researcher (GD) used to settle any disputes. In total, twenty-seven studies met the inclusion criteria and were included in the systematic review (See Fig 1).

## Data extraction

Data were extracted independently by two researchers (SC and AG) into a standardised spreadsheet, which included the characteristics of articles valid for review; including markers of gastrointestinal damage and permeability. Additional data were collected for study design; participant characteristics; the mode, volume and intensity of exercise; and training status of participants when provided. Due to recent findings where the timing of the saccharide absorption test drink (pre, during, post exercise) and fasted status prior to exercise (fasted/fed) had an influence on the urinary saccharide ratio, these data are noted in the summary table where necessary [10]. Supplement specific data were extracted as target nutrient, dosage and duration. Exercise data were extracted as mode, duration, intensity (percentage maximal oxygen uptake, percentage maximal power, time trial, time to exhaustion), and environmental conditions (temperature and humidity). Data from the change in biomarkers for each individual study are presented in the summary table (Table 1). These data were taken as reported from each study (mean±standard deviation / mean±standard error / median (interquartile range)), or where values were only presented in figure form, the figure was digitized using graph digitizer software (DigitizeIt, Germany) and the means and standard deviation/standard error or the median (interquartile range) were manually measured at the pixel level to the scale provided on the figure.

## Quality assessment of included studies

To assess the overall methodological quality of the studies, the Cochrane Collaboration tool for assessing the risk of bias was chosen [39, 40]. The tool targets areas of sequence generation,

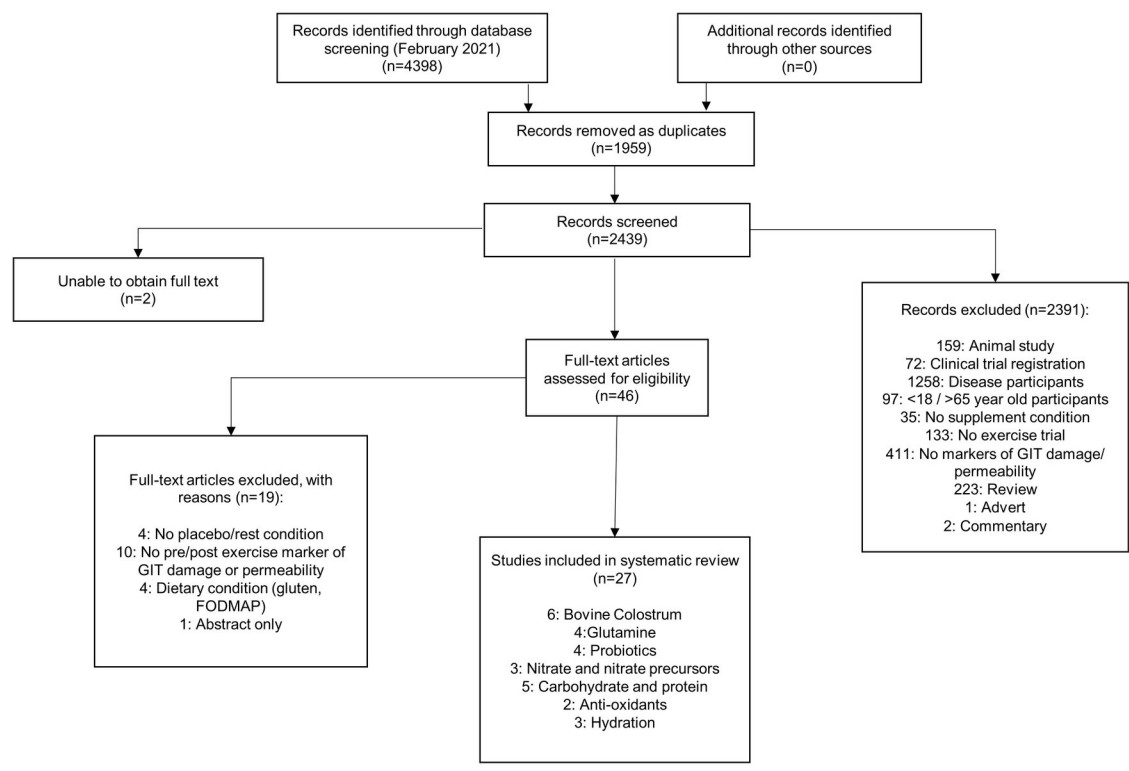

**Fig 1. Flow chart for selected studies according to the PRISMA guidelines.**

allocation sequence concealment, blinding of participants and researchers and outcomes reporting to evaluate possible sources of bias.

## Statistical analysis

Based on previous work, a high methodological heterogeneity between studies was expected with a limited number of studies investigating specific supplements [10]. Therefore, it was not deemed feasible to pool data for a meta-analysis for statistical comparison.

## Results

Twenty-seven studies were included in the systematic review (Fig 1), with a large heterogeneity in study design. Supplementation protocols ranged from a single dose pre and/or during exercise (n = 12, 24 hours pre-exercise) to multiple days of daily supplementation (n = 15; up to 8 weeks). Endurance-style exercise ranged from a 3000m running time trial and 20 minutes of treadmill running, to registered marathons and a four-day military hike. Nineteen studies used running as the mode of exercise. The majority of participants were trained male runners or cyclists with only 10% female participants. Thirteen studies were in heated environments (>23˚C) whilst the remainder were in thermoneutral conditions [41]. Intestinal fatty acid binding protein (i-FABP) was used as the primary marker of gut endothelial cell damage (n = 18), and twelve studies used a combination of urinary saccharides (Lactulose:Rhamnose (L:R)/ Lactulaose:Mannitol (L:M)/ Sucrose/ Sucralose) to evaluate different sites of intestinal permeability. Two studies used plasma saccharides measures [42, 43]. Supplements used in each of the studies are discussed separately below.

**Table 1. Summary data from studies investigating the impact of specific supplements on markers of gut cell damage and permeability.**

| Author | Participant details | Supplementation protocol (duration and dosage) | Exercise protocol | Outcomes measured | Result summary |
|---|---|---|---|---|---|
| **Bovine Colostrum (BC)** | | | | | |
| Davison et al. (2016) [5] | 8 active males (>4 training sessions/ week); 25 (19–33) years | **2 and 14 days** Placebo (capsule) + 20g placebo (powder) <br> **BC** (powder) 20g·day$^{-1}$ + placebo (capsule) <br> **ZnC** (75mg capsule) + placebo (powder) 20g·day$^{-1}$ <br> **ZnC** (75 mg capsule) + **BC** (powder) 20g·day$^{-1}$ | Fasted, Running for 20 mins at 80% VO$_{2(max)}$ in laboratory conditions | L:R[c] | **After 2 days:** **ΔL:R ↓ BC+ZnC vs placebo** ($p<0.01$) 0.023±0.002 → 0.045±0.002 (BC+ZnC) vs 0.023±0.002 → 0.063±0.002 **ΔL:R ↔ BC, ZnC vs placebo** ($p>0.05$) **After 14 days:** **ΔL:R ↓BC, ZnC and BC +ZnC vs placebo** ($p<0.01$) 0.023±0.002 → 0.045±0.002 (BC+ZnC) 0.023±0.002 → 0.043±0.003 (BC) 0.023±0.002→ 0.042±0.002 (ZnC) 0.023±0.002 → 0.065±0.004 |
| March et al. (2017) [46] | 18 recreationally active healthy males (26±5 years) | **14 days** **BC** (powder) 20g·day$^{-1}$ vs placebo (isoenergetic milk protein concentrate) | Fasted, Running for 20 mins at 80% VO$_{2(max)}$ at 22˚C /37% RH | L:R[c], i-FABP | **ΔL:R ↓ BC vs placebo (AUC)** ($p<0.001$) 0.34±0.05 → 0.50±0.06 vs. 0.35±0.06 → 0.95±0.15 **Δi-FABP↓ BC vs placebo** ($p = 0.013$) 672(394) → 684(481) vs 578(399) → 928(382 pg·ml$^{-1}$ |
| March et al. (2019) [44] | 12 recreationally active healthy males (26±6 years) | **14 days** **BC** (powder) 20g·day$^{-1}$ vs placebo (isoenergetic milk protein concentrate) | Fasted, Running for 60 mins at 70% VO$_{2(max)}$ at 30˚C /60% RH | i-FABP | **Δi-FABP↓ BC vs placebo** ($p = 0.015$) 727(682) → 1781(1603) vs 661(571) → 1924(1394) pg·ml$^{-1}$ |
| Marchbank et al. (2011) [45] | 12 recreationally trained males; 26 (19–38) years | **14 days** **BC** (powder) 20g·day$^{-1}$ vs placebo (isoenergetic milk protein concentrate) | Fasted, Running for 20 mins at 80% VO$_{2(max)}$ in laboratory conditions | L:R[c] | **ΔL:R ↓ BC vs placebo** ($p<0.001$) 0.022±0.00 → 0.026±0.00 vs. 0.023±0.00 → 0.042±0.00 |
| McKenna et al. (2017) [47] | 10 healthy active males (20 ± 2 years) | **14 days** **BC** (powder) 20g·day$^{-1}$ vs placebo (milk protein concentrate) | Fasted, Running 95% of ventilatory threshold (~46±7 mins) at 40˚C/ 50% RH | i-FABP | **Δi-FABP ↔ between conditions** ($p>0.05$) 989±490 → 1505±788 vs. 851±450 → 1267±521 pg·ml$^{-1}$ |
| Morrison et al. (2014) [38] | 7 trained (T) males (>6 training sessions/week) (23 ± 4 years) <br><br> 8 untrained (UT) males (<3 sessions/week) (21 ±2 years) | **7 days** **BC** (powder) 1.7g·kg·day$^{-1}$ vs placebo (corn flour) | Fed, Mixed mode–90 mins; 15 mins cycling (50% HRR), 60 mins running (30 mins 80% HRR and 30 mins distance), 15 mins cycling (50% HRR), at 30˚C /50% RH | i-FABP | **T: Δi-FABP ↔ between conditions** ($p>0.05$) 139±54 → 1110±480 vs 150±44 → 797±313 pg·ml$^{-1}$ <br><br> **UT: Δi-FABP ↔ between conditions** ($p>0.05$) 104±46 → 470±282 vs 211±103 → 428±266 pg·ml$^{-1}$ |
| **Glutamine** | | | | | |
| Osborne et al. (2019) [48] | 12 trained male cyclists (cycling a minimum of 2/week); 32±6 years | **Pre-exercise** **Glutamine** 0.9g·kg$^{-1}$ of FFM vs placebo (water and cordial) | 20km cycling TT at 35 ˚C/ 51% RH | i-FABP | **Δi-FABP ↔ between conditions** ($p>0.05$) 0.61(0.42–0.81) → 0.82(0.63–1.02) vs. 0.51(0.34–0.73) → 0.96(0.78–1.16) ng·ml$^{-1}$ |

(*Continued*)

**Table 1.** (Continued)

| Author | Participant details | Supplementation protocol (duration and dosage) | Exercise protocol | Outcomes measured | Result summary |
|---|---|---|---|---|---|
| Pugh et al. (2017) [42] | 10 recreationally active males; 24±4 years | **Pre-exercise** **Glutamine** 0.25; 0.5 and 0.9g·kg$^{-1}$ of FFM vs placebo (water and cordial) | Fasted, Running for 60 mins at 70% VO$_{2(max)}$ at 30°C /40-45% RH (10.1 ±0.9 km/hour) | Serum L:R[c], i-FABP | **ΔL:R ↓ glutamine vs placebo** 0.025±0.010 → 0.063±0.025 (0.25g·kg$^{-1}$) ES = 0.6; ± 0.5 0.025±0.010 → 0.067±0.027 (0.5g·kg$^{-1}$) ES = 0.5; ± 0.5 0.025±0.010 → 0.052±0.013 (0.9g·kg$^{-1}$) ES = 0.9; ± 0.6 vs.0.025±0.010 → 0.085 ±0.036 (placebo) **Δi-FABP ↓ glutamine (0.5 and 0.9g.kg$^{-1}$ of FFM) vs placebo** 313±146 → 596±304 (0.25g·kg$^{-1}$) ES = 0.02; ± 0.38 326±188 → 479±263 (0.5g·kg$^{-1}$) ES = 0.46; ± 0.54 262±183 → 486±226 (0.9g·kg$^{-1}$) ES = 0.44; ± 0.42 vs 351±204 → 600±240 pg·ml$^{-1}$ (placebo) |
| Zuhl et al. (2014) [50] | 8 endurance trained participants; (males = 5; females = 3), 25 ±4 years | **7 days** **Glutamine** 0.9g·kg$^{-1}$ of FFM·day$^{-1}$ vs placebo (lemon sugar-free drink) | Fasted, Running for 60 mins at 65–70% VO$_{2(max)}$ at 30°C and 12–20% RH | L:R[b] | **ΔL:R ↓ glutamine vs placebo** (p<0.05) 0.021±0.008 →0.027±0.007 vs 0.021±0.008 → 0.060±0.047 |
| Zuhl et al. (2015) [49] | 7 endurance trained participants; (males = 2; females = 5) 26 ± 4.4 years | **Pre-exercise** **Glutamine** 0.9 g·kg$^{-1}$ of FFM vs placebo (lemon sugar-free drink) | Fasted, Running for 60 mins at 65–70% VO$_{2(max)}$ at 30°C and 12–20% RH; 1,585 m altitude | L:R[b] | **ΔL:R ↓ glutamine vs placebo** (p<0.05) 0.02±0.01 → 0.04±0.02 vs 0.02 ±0.01 → 0.06±0.01 |
| | | | **Probiotics** | | |
| Axelrod et al. (2019) [51] | 7 endurance trained participants, 31 ± 2.3 years | **4 weeks** **Single strain probiotic** L. *Salivarius* UCC118; 2 x 10$^8$ CFU vs placebo (corn starch) | Running for 120 mins at 60% VO$_{2(max)}$ at 24°C/ 30.8%RH | L[b], R[b], Sucrose[b] | **ΔL ↔ between conditions (iAUC, relative to baseline)** (p>0.05) -0.07±0.2 vs 0.36±0.6 **ΔR ↔ between conditions (iAUC, relative to baseline)** (p>0.05) -0.05±0.2 vs 0.47±0.2 **ΔSucrose ↓ probiotic vs placebo (iAUC, relative to baseline)** (p = 0.029) -0.3±0.2 vs 1.6±0.7 (ug·kg$^{-1}$) |
| Mooren et al. (2020) [52] | 19 untrained male athletes (VO$_{2(max)}$ < 53 ml/kg min,18–35 years | **4 weeks** **Single strain probiotic** Escherichia coli strain *Nissle* 1917 (suspension, 5ml·day)* vs pre-supplementation trial | Running 60 mins and 60% VO$_{2(max)}$ (25mins) and 80% VO$_{2(max)}$ (25 mins) in laboratory conditions | i-FABP | **Δi-FABP ↓ in probiotic vs pre-supplement** (p = 0.037) 390±524→509±456 vs 384±450→ 559±465 pg·ml$^{-1}$ |
| Pugh et al. (2019) [43] | 24 recreational runners, (males = 20; females = 4), 22–50 years | **4 weeks** **Four strain probiotic (see below)** (n = 12) vs placebo (corn starch) (n = 12) **During exercise** Carbohydrate gels (66g/hour) with water (600ml/hour) during marathon | Fed, Simulated marathon (42.2km) outside (track); 16–17 °C. | Serum L:R[c] i-FABP | **ΔL:R ↔ between conditions** (p>0.05) 0.057±0.022 → 0.099±0.062 vs 0.061±0.042 → 0.081±0.036 **Δi-FABP ↔ between conditions** (p>0.05) 455±190 → 1814±1708 vs 460±221 → 1392±867 pg·ml$^{-1}$ |

(*Continued*)

**Table 1.** (Continued)

| Author | Participant details | Supplementation protocol (duration and dosage) | Exercise protocol | Outcomes measured | Result summary |
|---|---|---|---|---|---|
| Pugh et al. (2020) [31] | 7 endurance trained male cyclists, 23±4 years | **4 weeks** **Four strain probiotic (see below)** vs placebo (corn starch) **During exercise** 10% maltodextrin (8 ml·kg$^{-1}$ bolus and 2 ml·kg$^{-1}$ every 15 min, ~90g/hour) | Fasted, Cycling for 120mins 55% $W_{(max)}$ + 100kJ of work (simulated final sprint) in laboratory conditions | i-FABP | **Δi-FABP ↔ between conditions** (p = 0.374) 542±145 →342±192 vs 643±243 → 350±76 |
| Macronutrient | | | | | |
| Flood et al. (2020) [32] | 14 endurance trained participants; (males = 7; females = 7); 27 ± 8 years | **During exercise only** **Carbohydrate fluid** (16%) maltodextrin + fructose (M/F) (143ml/15mins, ~91g/cho·hour$^{-1}$) vs placebo (water) | Cycling at 45% $VO_{2(max)}$ for 90mins at 32˚C/ 70% RH with a TT (100 kJ of work) | i-FABP (n = 13) | **Δi-FABP ↓ in both carbohydrate conditions vs placebo (fold change)** (p<0.05) 205±265 (M/F/Pec) and 144±381(M/F) vs 549±320 pg·ml$^{-1}$ |
| | | **Carbohydrate fluid** (16%) maltodextrin + fructose with **pectin alginate** (M/F/Pec) vs placebo (water) | | | |
| Jonvik et al. (2019) [53] | 16 well trained male cyclists (9±3 hours/week training); 28±7 years | **Pre-exercise (3 hours and 15 mins)** **Sucrose** 2 x 20g·dose$^{-1}$ vs placebo (NaCl) | Fasted, Cycling for 60 mins at 70% $W_{max}$ in laboratory conditions | i-FABP | **Δi-FABP ↓sucrose vs nitrate and placebo (AUC)** (p = 0.002) 57,270±77425 (sucrose) vs 125,106±83,591 (sodium nitrate) and 114,907±91527 (placebo) pg·ml$^{-1}$ |
| | | **Sodium Nitrate** (800mg $NO_3$) vs placebo (NaCl) | | | |
| Karl et al. (2017) [8] | 73 army soldier volunteers, (males = 71; females = 2) Placebo (n-18);19±2 years Carbohydrate (n = 27); 20±1 years Protein (n = 28); 20±1 years | **Four days** **Added carbohydrate** snack, (added 4.4MJ/ ~1000kcal·day$^{-1}$) vs placebo (standard diet:14.6 MJ/ ~3500 kcal·day$^{-1}$) | Fasted, 4 day arctic military training exercise, 51-km cross-country ski-march, 50:10-min work-to-rest ratio with a ~45-kg pack | Sucralose (n = 49) | **ΔS (% excretion) ↔ between conditions** (p>0.05) 2.0±0.63 → 3.42±2.0 (carbohydrate) and 1.68±1.1 → 2.32±1.9 (protein) vs 2.0±0.4 → 2.83±1.1 (standard diet) |
| | | **Added protein** snack, (added 4.4MJ/ ~1000kcal·day$^{-1}$ vs placebo (standard diet: 14.6 MJ/ ~3500kcal·day$^{-1}$) | | | |
| Ma et al. (2020) [54] | 7 recreational long distance male runners; 18.7 ± 1.7 years | **8 weeks** **Hyper-immunised milk (IMP) powder** (20g·day$^{-1}$) vs placebo (protein powder) | 3000m running TT on a track (no conditions reported) | Urinary i-FABP /Creatinine | **Δi-FABP/Cr ↓ in IMP vs placebo** (p = 0.039) 370±450→ 4845±2402 vs 205±206→ 7609±4507 |
| Sessions et al. (2016) [55] | 7 trained participants; (males = 5; females = 2) 24±5 years | **During exercise only** **Carbohydrate gel** (27g glucose: fructose 2:1 ratio) vs placebo (water) | Running at 70% $VO_{2(max)}$ for 60 mins at 30 ˚C/ 12–20% RH | i-FABP | **Δi-FABP ↑ in carbohydrate vs placebo** (p = 0.02) 261±106→524±381 vs 251±130→337±207 pg·ml$^{-1}$ |
| Snipe et al. (2017) [30] | 11 non heat acclimatised participants; (males = 6; females = 5), 31±5 years | **Pre-exercise and every 20 mins during exercise** **Glucose** fluid (6% solution) (15g·20mins$^{-1}$) vs placebo (water) | Fed, Running at 60% $VO_{2(max)}$ for 2 hours at 35˚C/ 27% RH | i-FABP | **Δi-FABP ↓ glucose and protein vs placebo (change from baseline)** (p<0.05) 0 → 130±56 (glucose) and 0→ 89±51 (protein) vs 0→898±142 pg·ml$^{-1}$ from baseline |
| | | **Whey protein** (6.4% solution) vs placebo (water) | | | |
| Other nutrients | | | | | |
| Buchman et al. (1999a) [56] | 23 males, marathon in the previous 12 months, 25–49 years | **14 days** **l-arginine** (A) (n = 13) 30g·day$^{-1}$ vs placebo (P) (glycine) (n = 10) | Marathon race (Houston Methodist Marathon), -2 ˚C at start; 272 ± 46 mins (A), 215± 28 (P) | L:M[c] | **ΔL:M ↔ between conditions** (p = 0.47) 0.06±0.11→0.03±0.02 vs 0.02±0.01→0.03±0.02 |
| Buchman et al. (1999b) [57] | 25 males, marathon in the previous 12 months, 25–49 years | **14 days** Vitamin E (**d-α-tocopherol,** 1000 IU ·day$^{-1}$) (n = 10) vs placebo (soya lecithin) (n = 15) | Marathon race (Houston-Tennaco Marathon) 2h43–5h28 mins | L:M[c] | **ΔL:R ↔ between conditions** (p>0.05) 0.03±0.01→0.04±0.02 vs 0.03±0.02 →0.07±0.10 |

(*Continued*)

**Table 1.** (Continued)

| Author | Participant details | Supplementation protocol (duration and dosage) | Exercise protocol | Outcomes measured | Result summary |
|---|---|---|---|---|---|
| Szymanski et al. (2018) [58] | 8 non-acclimatised participants; (males = 6; females = 2), 19±2 years | **3 days** **Curcumin** (capsules) (500mg·day$^{-1}$) vs placebo (capsules) | Fasted, Running for 60 mins at 66–68% $VO_{2(max)}$ at 36.8°C at 25–27% RH, 8.7±0.7km/ hour | i-FABP | **Δi-FABP ↓ curcumin vs placebo** (p = 0.047) 834±117→1310±200 vs 878±117→1650±320 |
| Van Wijck et al. (2014) [33] | 10 recreationally trained males; 24±1 years | **Pre-exercise** **L-citrulline** 10g·dose$^{-1}$ vs placebo (L-alanine) | Fasted, Cycling for 60 mins at 70% $W_{max}$ in laboratory conditions | i-FABP | **Δi-FABP ↓citrulline vs placebo (% change from baseline)** (p>0.05) 0→ 127±19% vs 0 → 155 ±22% **Δi-FABP ↓citrulline vs placebo AUC** -(p<0.01) 185±506 vs 1318±533 |
| | | **Water and carbohydrate** | | | |
| Costa et al. (2019) [29] | 11 trained competing male runners, 34 ± 11 years | **During exercise** **Water** (844 +/- 271 ml·hour$^{-1}$; 11 +/- 3.5ml·kg) vs placebo (no water) | Running for 120 mins at 70% $VO_{2(max)}$ at 24°C/ 46% RH | i-FABP | **Δi-FABP ↔ water vs placebo** (Δfrom pre to post) (p>0.05) 368±476 vs 538±426 pg·ml$^{-1}$ |
| Kartaram et al. (2018) [59] | 15 recreationally active male cyclists (24±2 years) | **24 hours prior:** **Water *ad libitum* vs 0.5l·day$^{-1}$ water** **During exercise** **Water** vs placebo (no water) | Fed, Cycling for 60 mins at 70% $W_{max}$ in laboratory conditions | i-FABP | **Δi-FABP ↓ water vs no water** (p<0.001) 742 (341) → 1262 (512) vs 689 (309) → 1559 (658) pg·ml$^{-1}$ |
| Lambert et al. (2008) [60] | 20 distance trained runners, (males = 11; females = 9) 22± 3 years | **During exercise** **Water** (3ml·kg·10mins$^{-1}$) vs placebo (no water) **And glucose fluid** (3ml·kg$^{-1}$·10 mins$^{-1}$) vs placebo (no water) | Fasted, Running for 60 mins at 70% $VO_{2(max)}$ in laboratory conditions | L:R[b], Sucrose | **ΔL:R ↓ in water and glucose vs no water** (p<0.008)0.03 (0.01–0.11) → 0.05(0.02–0.12) (water) 0.03(0.01–0.11) → 0.05(0.01–0.11) (glucose) 0.03(0.01–0.11) → 0.06(0.02–0.17) (no water) **ΔSucrose (% excretion) ↓ in water and glucose vs no water** (p<0.008) 0.03(0–0.19) → 0.05(0–0.77) (water) 0.03(0–0.19) → 0.05(0–0.63) (glucose) vs 0.03 (0–0.19) → 0.09(0–2.01) |

Data are presented as increase (↓), decrease (↑) or no change (↔) between the pre and post exercise biomarker measures in the experimental, then placebo condition; data are presented as mean±SD, mean±SEM, median(interquartile range), area under the curve (AUC), incremental area under the curve (iAUC), or change from baseline based on the data available from the selected study; Δ, change; ZnC, Zinc Carnosine; i-FABP, intestinal fatty acid binding protein; L, Lactulose; R, Rhamnose; M, Mannitol; S, Sucralose; [a,b,c] illustrates the timing of the dual saccharide absorption test drink [a], pre-exercise, [b], during exercise, [c], post-exercise; $VO_{2(max)}$, maximal oxygen uptake; $W_{max}$, maximal power output in Watts; mins, minutes; °C, degrees Centigrade; TEE, time to exhaustion; CFU, colony forming units; *, dosage not specified in the study; 4 strain probiotic: B. *bifidum* CU20, L. *acidophilus* CUL-60, L *acidophilus* CUL-21, B. *animalis* sub sp *lactis* CUL-34, 25 billion CFU; Fasted or Fed refers to status prior to exercise, p values are those reported in the studies, comparing the magnitude of change between experimental and control groups, P<0.05 was considered significant.

Overall, the risk of bias across the selected studies was considered low. Sequence generation, allocation concealment and blinding of participants were low risk across the majority of studies. Four of the twenty-seven studies used placebo-controlled or matched pairs designs (no crossover), while all other studies used randomised blinded crossover protocols. Five of the studies were registered as clinical trials, meaning that the risk of selective reporting for the remainder of studies was unclear. The risk of bias summary for all studies can be found in Fig 2, and the risk of bias for individual studies are presented in the supplementary data (S2 File).

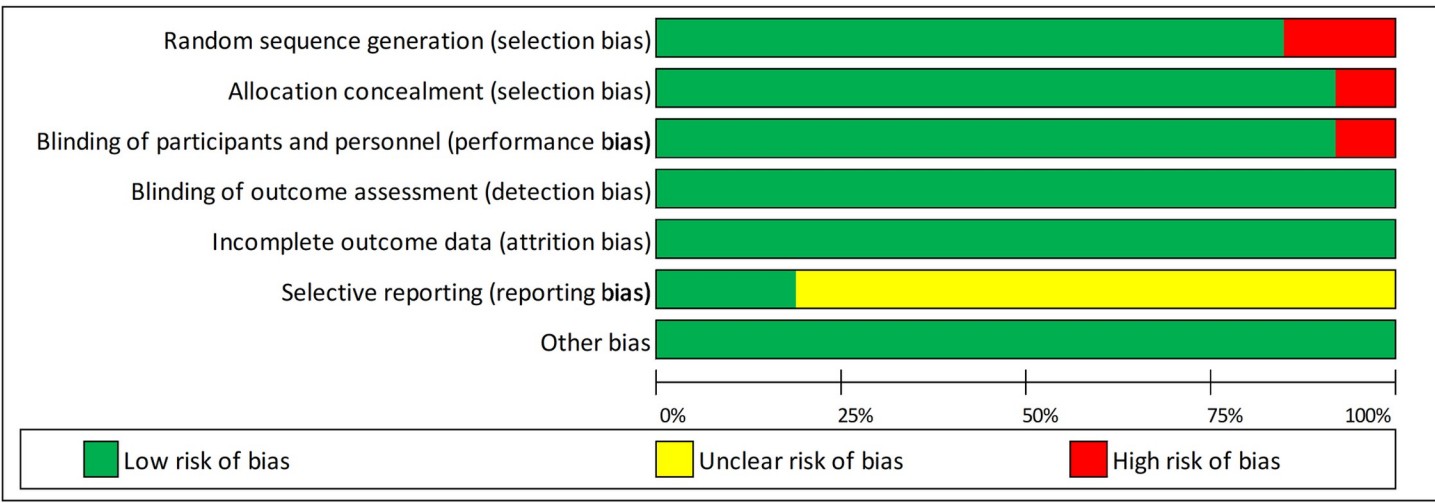

**Fig 2. Summary of the risk of bias for selected studies (released under creative commons CC BY SA Robin N. Kok v2.6 November 2017 www.robinkok.eu.**

## Bovine colostrum

Six studies investigated bovine colostrum supplementation, of which one included colostrum alone and in combination with zinc carnosine [5, 38, 44–47] (Table 1). Studies ranged from 2 to 14 days. Dosage-matched (20g·day$^{-1}$) studies evaluating endothelial damage (i-FABP) showed a positive blunting effect of colostrum subsequent to exercising at 22 ˚C [46] and 30˚C [44] but not at 40˚C [47]. Morrison et al (2014) showed a larger change from baseline to post-exercise (90mins) in i-FABP levels in trained compared to untrained participants, but showed no effect of colostrum, in spite of the high dosages given [38]. Permeability measures were lower in colostrum compared to a placebo in two studies in thermoneutral laboratory conditions [5, 45]. An additive benefit was found with the addition of zinc carnosine compared to the other experimental conditions with L:R being 85% lower than the placebo condition after 14 days of supplementation [5].

## Glutamine

Four studies investigated glutamine, with three showing lower markers of gut endothelial damage post-running in the glutamine condition [42, 48–50] (Table 1). All studies were performed in heated environments (30–35˚C). Three used a single dosage-matched protocol pre-exercise (0.9g·kg$^{-1}$) [42, 48, 49]. One study tested a graded dosage (0.25, 0.5 and 0.9g·kg$^{-1}$ of fat free mass (FFM)) prior to 60 minutes of heated running. Magnitude based inferences suggested a *likely* lower i-FABP levels post-exercise in the 0.5 and 0.9g·kg$^{-1}$ FFM conditions compared to the placebo [42]. In contrast, there was no difference in i-FABP levels post-20km cycling time trial between conditions (0.9g·kg$^{-1}$ FFM vs placebo) [48]. There were lower markers of permeability (urinary L:R) post-running after both an acute (pre-exercise) and chronic (7 days) supplementation protocols compared to a placebo (0.9g·kg·day$^{-1}$ FFM) [49, 50].

## Probiotics

Four studies examined the effects of different probiotic strains on markers of gut permeability and endothelial damage [31, 43, 51, 52] (Table 1). The studies that supplemented with a single-strain probiotic showed some benefit in gut cell damage [52] and proximal gut permeability

[51] (*Escherichia coli* and *Lactobacillus salivarius* respectively). In contrast, the studies using four-strain probiotic supplementation *(B. bifidum* CU20, *Lactobacillus acidophilus* CUL-60, *Lactobacillus acidophilus* CUL-21, *B. animalis* sub sp lactis CUL-34; 25 billion CFU) showed no difference between conditions. Of note, Pugh and colleagues included exogenous carbohydrate supplementation during the exercise due to the length of the exercise bout of ~66g·hr$^{-1}$ (repeated 22g maltodextrin gels with 200ml water during the running marathon) and ~90g·hr$^{-1}$ (10% maltodextrin fluid every 15 minutes during 120 minutes of cycling) in the four-strain probiotic studies [31, 43].

## Macronutrients

Seven studies compared carbohydrate and/or protein-based foods and fluids, to a placebo [8, 30, 32, 53–55, 60] (Table 1). A single study found an increase in i-FABP in response to carbohydrate (27g) supplementation during a cycling time trial [55], while another found no impact of carbohydrate fluid compared to a placebo post-run [60]. Four studies using carbohydrate fluids (of various concentrations; 6–16% solution) mitigated the impact of exercise with lower i-FABP concentrations [30, 32, 53, 60] although all used different saccharide combinations and concentrations limiting direct comparison (specific values in Table 1).

Two studies considered the impact of acute and chronic protein supplementation, albeit with different protein compositions. Whey protein (6.4% solution) consumed during exercise resulted in lower markers of gut cell damage and permeability compared to water [30]. Eight weeks of hyperimmunised milk protein supplementation (20g·day$^{-1}$) mitigated endothelial cell damage (urinary i-FABP/creatinine) in response to a 3000m time trial compared to whey protein (placebo) [54]. However, additional carbohydrate or protein as snacks (~1000kcal·day$^{-1}$) over a four day military hike in the arctic had no impact on intestinal permeability markers (sucralose excretion) compared to a control group (no snacks) [8].

## Anti-oxidants, nitrate and nitrate precursors

Two studies investigated the impact of anti-oxidants [57, 58] (Table 1). Three days of curcumin supplementation (500mg·day$^{-1}$) significantly reduced the impact of 60 minutes of running in the heat on i-FABP levels [58]. In contrast, Vitamin E (1000 IU d-α-tocopherol·day$^{-1}$) supplementation for 14 days had no impact on permeability (L:M) post-marathon [57].

Three studies evaluated the impact of dietary nitrate or nitric oxide precursors targeting nitric oxide (NO) availability [33, 53, 56] (Table 1). A single dose of sodium nitrate did not affect i-FABP compared with placebo post-exercise [53]. L-citrulline pre-exercise, as an arginine precursor, showed lower concentrations of i-FABP (measured every 15 minutes) and area under the curve during exercise compared to placebo, in spite of similar changes from baseline to post-exercise (72% increase) [33]. L-arginine supplementation over 14 days made no impact post-marathon on gut permeability markers [56].

## Hydration status

Three studies investigated the role of hydration status. Two studies evaluated the impact of complete fluid restriction during exercise and showed higher markers of gut cell damage and permeability (i-FABP, L:R) compared to water *ad libirtum* [29, 59]. The final study showed higher permeability of the upper gastrointestinal tract (Sucrose) with fluid restriction compared to water or carbohydrate fluid [60] (Table 1). However, there was no significant difference between the carbohydrate and water conditions.

## Discussion

The aim of this systematic review was to synthesise the current available data on the impact of acute or chronic dietary supplement protocols on markers of gut endothelial cell damage and permeability in response to exercise. Studies showed that bovine colostrum and glutamine may be useful in moderately heated (<30˚C) endurance environments. Carbohydrate supplementation, specifically maltodextrin (glucose) and fructose, prior to and during exercise reduced endothelial cell damage compared to water, but there is no clarity around optimal dosage or saccharide type. Water intake, beyond being used as a placebo, showed lower markers of cell damage compared to fluid restriction. Probiotics require further research around dosage and strain specificity for gut specific outcomes. Other supplements, such as nitrate, nitrate precursors or anti-oxidants demonstrated equivocal findings and need more research to support any inclusion in an athlete nutrition plan. In summary, no supplement showed results that can be generalised across athletes, and therefore, practitioners and athletes should introduce with caution.

### Bovine colostrum

The bioactive compounds present in bovine colostrum are hypothesised to improve gut cell wall integrity as well as tolerance to hypoxia, thermal, and oxidative stress [45]. Four of six studies presented here showed positive effects. It is one of the few supplements to draw from both *in vivo* and *in vitro* studies, informing and improving the understanding of the physiological mechanisms. *In vitro*, colostrum supplementation, and colostrum combined with zinc carnosine, protect against temperature induced increases of apoptosis and upregulated heat shock protein induction and the phosphorylation of tight junction proteins in human intestinal cell lines [5, 45]. These mechanisms may be similar and therefore contribute to the results seen *in vivo* with improve thermotolerance of tight junctions in response to exercise and hence the lower i-FABP and dual saccharide levels seen in the *in vivo* studies [5, 44, 45]. Some of the mechanisms have been investigated and reviewed [61] but will require further research around exercise to elucidate further.

Supplementation of zinc carnosine and bovine colostrum showed additional benefit compared to colostrum or zinc alone [5]. The protocol showed a synergistic effect as the combination reduced the exercise-associated intestinal permeability significantly more than bovine colostrum alone, or zinc carnosine alone [5]. Zinc carnosine has previously been shown to decrease NSAID-associated gut permeability [62]. This compliments existing research showing an important role for zinc in immune function and the health of the gut mucosal membrane [63]. Often clinical studies do not test possible synergistic effects, but this was demonstrated by the four way double blinded randomised cross over study [5]. This is the only study to have evaluated Zinc in this context.

In spite of the improved thermotolerance indicated by the *in vitro* studies, one study showed no benefit of colostrum supplementation (20g·day⁻¹) on markers of gut damage (i-FABP) post-running at 40˚C [47]. Heat, in addition to exercise, amplifies splanchnic hypoxia by redistributing blood to the skin, possibly contributing to additional endothelial cell damage [12]. This has been shown with significantly higher i-FABP concentrations post-exercise in heated compared to thermoneutral environments [10]. While initial colostrum data supports improved thermotolerance at 30˚C [44], this effect may be negated at higher temperatures. Another reason for the difference in results may be a difference in the bioactive components in the bovine colostrum itself. Due to some inconsistent results between bovine colostrum studies, Halasa and colleagues investigated the impact of delayed milking post-partum on the ability of bovine colostrum to reduce permeability markers in recreational athletes [64]. They

found that bovine colostrum obtained 72-hour post-partum was less effective than colostrum from 2- and 24-hours post-partum. They surmised that the difference in efficacy may be from the natural change in bioactive compounds post-partum. This has been confirmed by recent laboratory studies [65]. Studies here used products that claimed milking within the first day (24 hours) post-partum [5, 44, 47], but not all studies had the information available, and this may be a factor.

In the single study that considered trained and untrained participants, training status did not alter the response to bovine colostrum supplementation compared to the placebo condition. The authors did note a higher i-FABP concentration post-exercise in the trained participants compared to untrained (p = 0.006), but this was independent of colostrum supplementation [38]. Higher output during exercise based on training status reflected in body temperature, could have contributed to this difference, but there was no significant difference in body temperature during exercise between groups. All, but one study [52], used trained participants, and therefore, there is limited understanding if training status alters the gut perfusion-exercise relationship. This would be useful for future studies to consider. This study used a high dosage of bovine colostrum (126-140g·day$^{-1}$, based on mean body mass) with no reported negative effects [38]. However, as there were no measurable benefits to this high dosage, 20g·day$^{-1}$ may be an appropriate dosage based on the findings from other studies.

## Glutamine

Glutamine supplementation had a positive impact on markers of gut damage and permeability in both the acute and chronic supplementation studies using running [42, 49, 50], but not cycling [48]. Pugh et al (2017) illustrated a possible dose-dependent benefit but was the only study to compare different dosages. Higher dosages (0.9g·kg) had minimal impact on GIS in spite of some concerns around tolerance [42]. While there are no studies directly comparing markers of gut damage induced by different exercise modalities, certain studies have shown that subjective (self-reported gastrointestinal symptoms) and objective (gastroesophageal reflux) gastrointestinal issues are higher in running compared to cycling [66, 67]. This has been partially attributed to differences in biomechanical movement, as well as the possible implications of drinking or eating on breathing patterns whilst running [66]. Whilst elevated i-FABP levels were observed in both groups post-cycling, the authors reported no increase in endotoxin levels. This may illustrate initial endothelial cell damage but without sufficient physiological stress to cause increased permeability and bacterial translocation, suggesting the exercise stimulus was not sufficient to benefit from higher glutamine availability [48].

It is worth noting that in a separate study, no additional benefit to permeability measures was found with adding glutamine (0.6%) to a carbohydrate drink (6%) after 60 minutes of running [68]. This aligns with data in rats to suggest that in the presence of glucose, both substrates can improve energy production for the enterocytes [69]. Glutamine is considered conditionally essential under physiological stress as a primary fuel for gut mucosal cells as well as a being a crucial component of immune cells [70]. *In vitro* data by Zuhl et al. (2014) showed glutamine supplementation augmented heat shock protein 70 (HSP70) and heat shock factor-1 (HSF-1) expression with heat exposure, alongside preserved occludin expression [50]. Heat shock protein upregulation can improve cell tolerance to oxidative stress and inflammation and interact with occludin to preserve tight junction integrity [71]. This is reflected in the lower permeability markers (L:R) levels seen by Zuhl and colleagues [49, 50], and may highlight multiple modes of action where glutamine can improve tight junction stability and increase thermotolerance [72]. Therefore, based on the included studies, glutamine may be most useful if carbohydrate is restricted to preserve gut endothelial function in the heat in

runners. To note, chronic high dosages are not recommended due to possible alterations in amino acid metabolism [73].

## Probiotics

Of the four studies in this review, there were positive results with lower upper gastrointestinal permeability and endothelial cell damage post-running, but only in single-strain protocols [51, 52]. The two single strains (*Escherichia coli* and *Lactobacillus salivarius)* have been shown to improve tight junction stability in response to induced damage *in vitro* [74, 75]. Other sub-species of *Lactobacillus* were used in the four-strain studies (L. *acidophilus*; CUL60 and 61) without any impact on endothelial damage and permeability. This aligns with the current research showing that effects of probiotics are dosage and strain specific [76]. Probiotics reportedly contribute to an improved microbiota profile which one might expect to increase exercise tolerance of the gut endothelium [77, 78]. Unfortunately only one study presented here examined the change in microbiota, where they showed some minor changes and no shift in diversity measures [51], limiting the understanding of the knock-on effects of the strains.

Furthermore, two studies gave supplemental carbohydrate during the exercise trial due to the duration (≤120mins) [31, 43]. This may have altered the ability to differentiate between the two groups, based on the effect of exogenous carbohydrate [30]. Alongside, one study showed no exercise-associated increase in markers of gut damage (i-FABP), which reflected the low intensity of the exercise protocol (55% maximal power output) [31]. This same study did report higher exogenous carbohydrate oxidation in the probiotic group compared to placebo [31], while the other reported lower gastrointestinal symptoms during the last third of the marathon [43]. This may illustrate that any benefits from probiotics for athletes may come via different mechanisms, which have been illustrated in other studies investigating illness severity [79, 80] or overall performance [78, 81].

As mentioned, Jäger et al (2019) showed that the results of probiotic use are influenced by species, dosage and supplementation duration [82], which adds to the complexity of any practical prescription for athletes. While four weeks of supplementation may be long enough to see changes, the gut microbiome is highly individualised, and responsive to intervention [83]. While it is not practical to control dietary intake for long periods of time, a recent study showed that a three-week dietary intervention was associated to measurable changes in microbiome and metabolites in elite race walkers during a training camp [84]. In addition, there is an independent effect of exercise and exercise volume (specifically endurance) on gut microbiota diversity and species profiles [85–89]. Longer duration events (days) have been shown to alter the gut microbiome profile and resulting metabolites [8, 89]. Further discussion is beyond the scope of this review, but the individual variability and flux in dietary intake and training regimes complicates results found in probiotic studies. Overall, different probiotics may improve the microbiota profile or endothelial cell integrity and therefore the gastrointestinal tolerance to a 'stressful' training/race period [8], but more research is required to establish the mechanisms whereby we see some positive impact.

## Macronutrients

Despite the well-established ergogenic effect of carbohydrate supplementation on endurance performance [90, 91], research exists demonstrating that different saccharides (e.g. glucose, fructose) can impact gastrointestinal comfort [92] and are popular amongst self-selected athlete dietary exclusions [93]. While Sessions et al. (2016) did find that a mixed glucose/fructose gel (27g) during exercise increased gut cell damage (i-FABP) compared to water [55], other studies presented here found either negligible impact [8, 60] or a positive

blunting in markers of gut damage with carbohydrate supplementation [30, 32, 53]. The form of the carbohydrate gel (semi-solid) or the high concentration of the bolus may have affected gut damage (27g/250ml water, 10.6%), but this result seems contrary to the other studies that have used carbohydrate gels [43]. The range of carbohydrate intakes (~20-90g·hr$^{-1}$) and use of different saccharide combinations makes any optimal dosage or combination challenging to determine.

In spite of the carbohydrate intake variability, the continuous provision of exogenous fuel (glucose, fructose or sucrose) may improve the energy availability to gut cells, limiting the exercise associated damage, especially in the heat [30, 53]. Based on these results above, it is interesting that there was no impact of extra carbohydrate or protein snacks (~1000kcal·day$^{-1}$) on permeability markers during a four-day military hike [8]. It was noted that the participants had an average energy deficit of 55% compared to their energy expenditure (and a 2.7±2.1kg loss in body mass). This may indicate that an acute energy deficit may prioritise macronutrients elsewhere and decrease energy available to the endothelial lining [8]. This adds to the estimated high metabolic requirements of the gastrointestinal organs. Research in piglets has shown that ~50% of dietary amino acids are used by the gastrointestinal organs (intestines, stomach, spleen and pancreas) after feeding [94]. These gastrointestinal associated energy costs may be higher in athletes over longer duration exercise. Considering energy balance, gastrointestinal symptoms and illness have been found to be higher in athletes with low energy availability, specifically females [95, 96], which may align with these findings.

Methodologically, another reason for the variation in results may be the timing (pre and/or during) of carbohydrate-containing fluid and being fed/fasted prior to exercise. Permeability markers (L:R) have been shown to be lower in studies where the participants were fed compared to fasted prior to exercise [10]. Pre-exercise feeding (containing carbohydrate and/or protein) may reduce the magnitude of the splanchnic hypoperfusion at the beginning of exercise due to digestion requirements. This trend was shown previously with a maintenance of portal perfusion with carbohydrate compared to water during running [97]. This may explain the lack of difference in permeability in some studies starting in a fed state [98], but is likely to interact with the factors such as exercise duration, intensity and temperature. Many of the studies target a low FODMAP pre-exercise meal, but there is also no understanding as yet of the impact of fibre (in solid or fluid) intake prior to exercise on gastric emptying and endothelial cell support.

Acute protein supplementation had a positive impact on markers of gut damage [30]. Although whey protein did blunt the change in i-FABP levels in response to exercise, the authors reported higher gastrointestinal symptoms in the protein trial, which may therefore be impractical [30]. The authors saw similar changes in i-FABP with carbohydrate supplementation and concluded that the blood flow to the microvilli was sustained with macronutrient intake during exercise. Chronic supplementation of concentrated hyperimmunised milk powder resulted in lower i-FABP levels in the urine [54]. There is limited data around using hyperimmunised milk powder, and how this compares to the bioactive profile of bovine colostrum. The rationale of the study was that the additional immune properties would be more beneficial than protein powder supplementation alone. While the study showed a benefit in lower i-FABP levels post 3000m time trial (~9–10 minutes) compared to standard protein supplementation, the study used urinary i-FABP as the marker of endothelial damage adjusted for urinary creatinine levels. This may highlight renal clearance of i-FABP rather than systemic appearance and may require prudent interpretation. This measure has yet to be validated elsewhere and may contribute to the >10 fold increase in concentration.

## Nitrate and nitrate precursors

Arginine, citrulline and nitrate supplementation are hypothesised to increase nitric oxide (NO) availability via the nitrate-nitrite pathway, and improve vasodilation and tolerance to hypoxia during exercise [99]. This could be beneficial if it extended to the perfusion of the gut. The effect of dietary nitrates and NO precursors on performance have been reviewed elsewhere [100–102]. In this review, one study showed no benefit of arginine supplementation for attenuating exercise-induced gut cell damage and permeability [56]. L-citrulline is the NO precursor of choice due to its improved bioavailability compared to arginine [101]. Citrulline supplementation resulted in a lower plasma i-FABP concentration (area under the curve) after an hour of cycling [33]. Increases in plasma citrulline concentration post supplementation corresponded to a maintenance of splanchnic perfusion ($gapCO_2$) during exercise, and lower i-FABP concentration, illustrating a possible NO-mediated effect [33]. Conversely, sodium nitrate supplementation increased plasma nitrate concentration but did not impact $gapCO_2$ levels, and had no consequent benefit to endothelial damage [53]. Although the dosage of sodium nitrate (800mg nitrate) was matched to other studies, it may have been affected by the acid reducing medication taken prior to the study to assist with the $gapCO_2$ measurement, since an acidic environment is preferred for the nitrite conversion. This would explain the increase in plasma concentration with no further benefit. Recently, the role of the oral microbiome as an important adjunct for the conversion to nitrite was confirmed and may be affected by macronutrient balance in an athlete's diet [102–104]. Practically, a diet rich in nitrate containing foods can achieve the same plasma nitrate concentration as supplementation [105]. Nitrate supplementation (via beetroot and/ or other high nitrate foods, e.g. rhubarb and amaranth) has become popular with athletes and more research to investigate the perfusion sparing effect to assist with gut related ischaemia is needed.

## Antioxidants and other nutrients

A moderate dosage of curcumin showed capacity to reduce endothelial cell damage [58], while vitamin E had no impact [57]. The limited number of studies using antioxidants makes it difficult to draw specific conclusions. Curcumin, as a part of turmeric and curry powders, has been shown to promote tight junction stability and may have a role in microbial diversity [106, 107]. This informs the lower i-FABP concentrations, and inflammatory cytokines seen after heated (~37˚C) running compared to a placebo shown in the selected study [58]. The authors suggested that less damage to the intestinal barrier might have reduced bacterial translocation, and therefore the need for an inflammatory response. These results were accompanied by lower physiological strain and change in core temperature compared to the placebo [58]. Other positive findings around curcumin and gut derived effects are mostly *in vitro* and studies struggle to be replicated *in vivo* due to its low bioavailability [108]. There is currently an effort to improve the bioavailability of curcumin and initial results in intestinal disease are promising [109]. Further studies in different sporting populations, different durations and replicability are required to increase the understanding of this popular compound.

Vitamin E supplementation had no benefit on gut permeability in marathon runners [57]. The large range of finishing times, reflecting a heterogeneous sample of runners, may have influenced the result, but there is limited data to support that anti-oxidants can be effective in a localised manner on the gut endothelial lining. Foods and spices (e.g. plant polyphenols, curcumin, pomegranate, tart cherry, grape extract) that have anti-oxidant properties may target multiple mechanisms for reducing oxidative damage or improving redox balance but

performance and immune function studies still support food-based intakes of fruit and vegetables rather than supplementation [110, 111]. Chronic supplementation with high doses of anti-oxidants is not currently recommended due to possible dampening of other physiological adaptions associated with some types of training [24, 112], and therefore, further research will be needed to confirm their place amongst nutrition strategies for gut health.

## Hydration

In the included studies there was a consistent finding illustrating the negative impact of water restriction during exercise on markers of gut cell damage and permeability [29, 59, 60]. Fluid restriction may exacerbate the impact of splanchnic hypoperfusion, ischemia and increased the risk of hyperthermia, causing tight junction disruption and increased endotoxaemia [11]. Studies included here showed increased plasma i-FABP concentrations [59] and permeability measures [60] when fluid was restricted (~1.5% body mass loss) during exercise. Further fluid restriction (resulting in a ~3.1% loss in body mass) did not exacerbate this response [29]. Due to the possible negative impact of large fluid losses (>2–5% body mass) on exercise performance, the included studies continue to support that hydration in hot or long duration events is important and may improve gastrointestinal outcomes along with other components of performance. The impact of hydration status on marker of gut damage should be kept in mind for research when using water as a placebo condition. In addition, the temperature of fluid may play a role as previous research indicated that cold water (0.4 ± 0.4˚C) can also mitigate gut damage after 2 hours of running in the heat (35˚C) and improve upper GIS compared to cool water (7.3 ± 0.8˚C) and water at room temperature [113].

## Limitations and future research

Whilst this review provides an overview of the current literature specific to markers of gut cell damage and permeability, there are several limitations associated with research of this nature. The biomarkers of gut cell damage and permeability used in athletes are adapted from clinical studies, without consensus on clinically relevant cut-offs. Although supplement strategies may improve damage at the endothelial cell level, the translation to improve GIS and long-term GI health needs further elucidation. The markers are valid and reliable in athlete populations, but there are still no standardised protocols [4, 21, 114–116]. Introducing a standardised protocol for DSAT would be an advantage since there were studies excluded from this review due to no resting control. Longitudinal studies cannot fully control for dietary intake or training programmes during chronic supplementation periods, making it difficult to isolate specific effects. While the risk of bias was low and studies were individually well designed, the large heterogeneity in experimental design (even within supplements) affects the ability to generalise results. In the broader context of gastrointestinal distress alongside supplements, 'gut training' [117, 118] and dietary interventions such as low FODMAP diets [26, 27, 119] have also had positive results and have been recommended to limit the impact of exercise associated damage. Finally, there are no studies in team or intermittent sports, nor equal representation of female athletes. Collision sport such as Rugby League have shown higher metabolic cost post-collision [120], and further investigation into the mechanical impact on the gut may be warranted, especially with anecdotal reports of high non-steroidal anti-inflammatory usage. While there is no clear sex-based differences in response to exercise expected [37], other issues such as higher rates of irritable bowel syndrome in females and menstrual cycle associated changes in GIS deserve more attention [121]. As such, a greater quantity of research should be conducted which relates to specific athletic populations and sports.

## Conclusion

This systematic review identified specific supplements that may benefit the initial phase of exercise-associated gastrointestinal syndrome. Bovine colostrum, with or without zinc carnosine, and glutamine may ameliorate exercise induced gut damage and permeability in running, especially in moderate heat and if carbohydrate is restricted. Maintaining euhydration and consuming exogenous carbohydrates are likely to blunt the response to exercise. Antioxidants and nitrate or nitrate precursors require more research to confirm the mechanisms of action and their place within nutrition strategies. Probiotics may have a role within the larger picture of gastrointestinal health, but the mechanisms are not evident. As such, many of the supplements presented here should be evaluated for their effectiveness in a context dependent manner. External factors such as previous GIS, exercise modality, duration and intra-event fuelling opportunities should contribute to decision making. These data provide clarity on the current literature and may be useful to inform nutritional strategies to attenuate gut damage and permeability in response to exercise.

## Supporting information

**S1 File. Search strategy for systematic review.**
(DOCX)

**S2 File. Summary of the risk of bias for individual studies.**
(DOCX)

**S3 File. PRISMA checklist.**
(DOCX)

## Author Contributions

**Conceptualization:** Sarah Chantler, Alex Griffiths, Glen Davison, Adrian Holliday, Ben Jones.

**Data curation:** Jamie Matu.

**Methodology:** Sarah Chantler, Alex Griffiths, Jamie Matu, Glen Davison.

**Supervision:** Glen Davison, Adrian Holliday, Ben Jones.

**Writing – original draft:** Sarah Chantler.

**Writing – review & editing:** Sarah Chantler, Alex Griffiths, Jamie Matu, Glen Davison, Adrian Holliday, Ben Jones.

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
