## [Decision Letter · Decision Letter 0]

2 Jun 2021

PONE-D-21-15349

A SYSTEMATIC REVIEW: ROLE OF DIETARY SUPPLEMENTS ON MARKERS OF EXERCISE-ASSOCIATED GUT DAMAGE AND PERMEABILITY

PLOS ONE

Dear Dr. Chantler,

Thank you for submitting your manuscript to PLOS ONE. After careful consideration, we feel that it has merit but does not fully meet PLOS ONE’s publication criteria as it currently stands. Therefore, we invite you to submit a revised version of the manuscript that addresses the points raised by the reviewers during the review process.

We look forward to receiving your revised manuscript.

Kind regards,

Pedro Tauler, Ph.D.

Academic Editor

PLOS ONE

Journal Requirements:

5. Thank you for stating the following in the Financial Disclosure section:

We note that one or more of the authors are employed by a commercial company: Leeds Rhinos Rugby League Club, Rugby Football League and Yorkshire Carnegie Rugby Union Club

Reviewers' comments:

Reviewer's Responses to Questions

**Comments to the Author**

1. Is the manuscript technically sound, and do the data support the conclusions?

Reviewer #1: Partly

Reviewer #2: Yes

2. Has the statistical analysis been performed appropriately and rigorously? 

Reviewer #1: N/A

Reviewer #2: N/A

3. Have the authors made all data underlying the findings in their manuscript fully available?

Reviewer #1: Yes

Reviewer #2: Yes

4. Is the manuscript presented in an intelligible fashion and written in standard English?

Reviewer #1: Yes

Reviewer #2: Yes

5. Review Comments to the Author

Reviewer #1: Dear author,

Thanks for submitting your manuscript. See below for my comments.

Overall comments:

It is quite hard to claim an effect on such a multifactorial problem.

There is no causal cause between GIS and specific gut damage markers.

Methods of included studies can be described more indept to increase clarity.

With high carbohydrate intakes we often see fructose-intolerances (or even fructose allergics) which also leads to GIS: how are you sure about excluding these individuals?

L77-89: The pathophysiology is poorly described. Please take a look at papers like de Oliveira et al 2014 and Costa et al 2016 and rewrite this specific part.

L158-159: Is the DSAT validated? If so, what are the pro and cons of this method?

L224-225: i-FABP post exercise is not a reliable marker. You need te pre exercise value to get insights in the increase of i-FABP. Please come back to this in the discussion part.

L260-263: What was in the CHO supplement? The effect of glucose for example is different that the effect of a olygosacharide.

L278: there are different kind of measurements taken (blood i-FABP and urinary-i-FABP) which makes the results very questionable.

L305-311: Rewrite this please. However the risk of bias is low, it has to be undelined that the outcomes are based on specific circumstances.

L317-319: Please dig deeper upon the saccharide types.

L383-384: I would like to see a discourage of usage because of the mentoined reasons.

L409-410: Why is only usage in high temperature highlighted? There are multiple practical implications you could think of.

L482-490: What about fiber intake?

L614-618: the conclusion is missing the specific environmental settings. Please expend 'context dependent' with some of the factors that may influence GIS.

Reviewer #2: The authors provide a systematic review of a variety of dietary supplements and their impact on exercise-associated gut damage and gut permeability, primarily measured via Intestinal fatty acid binding protein (i-FABP) and/or dual saccharide absorption tests. The topic is of interested and the review methodology is strong following the PRISMA guidelines. A few suggestions are provided by section below.

Introduction

Line 86- It would be nice to have more clarity regarding specifically, which gastrointestinal symptoms are associated with increased endotoxin levels, as well as, an estimated time to on-set and duration of the effects.

Line 88 – Changes in gastric emptying and motility may affect nutrient absorption. Consider discussing, as this may affect gut permeability and be relevant to the saccharide absorption tests.

Line 99 – Given that some of the studies use an acute supplementation and chronic, it would be valuable to discuss potential mechanism and/ore effects of both. For instance how would an chronic supplementation impact gut permeability during a bout of exercise vs. an acute protocol.

Methods

Why were your outcome measures not included in your “key word” search list?

Line 138 – It would help the reader to clarify that the ratio of lactulose to rhamnose/ mannose is used and that an increase in lactulose indicates increased permeability.

Line 164 – inclusion criteria for the exercise would be valuable if there were any. i.e. endurance exercise etc. It appears there are no studies looking at strength or intermittent exercise. Note if they were specifically excluded or simply not available. I see there is some discussion on this in the limitations with respect to “team” sports. Finally, were there any criteria regarding the amount of exercise pre-test day i.e. after a “rest day” or dietary controls i.e. “no caffeine” or drug controls i.e. “no NSAIDS”?

Results

Table 1 is comprehensive and well done. Table description for clarity confirm that difference is significant and p-value used.

Table 1 – last row/column should “an” be “and”?

Line 281 - Macronutrients can any statements be made regarding liquid vs solid based on the data?

Discussion

Bovine Colostrum – some insight into potential mechanism would be of interest if available. The in vitro studies do mention potential protective measures, but how these could be occurring is not discussed.

Line 356 – “affect” should be “effect”

It is odd that in line 305 it was noted that subgroup analysis could not occur and yet later it is noted that training status did not alter the response (line 368). I’m not sure there is sufficient data here to support the statement.

Line 383 “it’s” should be “its”

Consider expanding the discussion on heat shock protein and occludin to relate back to the current studies. One or two sentences to explain to the reader how these affect gut permeability should be sufficient.

Line 574 – “marker” should be “markers”

Limitations

Potentially lack of controls in the study regarding caffeine and NSAIDs as mentioned in methods sections.

Line 597 – I also didn’t see studies in strength based or HITT/cross-fit type workouts

Line 601 – females generally experience higher levels of exercise-induced GI symptoms than males.

6. PLOS authors have the option to publish the peer review history of their article (what does this mean?). If published, this will include your full peer review and any attached files.

Reviewer #1: **Yes: **Daan Hoogervorst

Reviewer #2: No

---

## [Author Response · Author response to Decision Letter 0]

17 Aug 2021

Response to the reviewers: 

Thank you for taking the time to review the manuscript. We have aimed to address your concerns below, and hope you will find our changes to your satisfaction. 

Reviewer #1: Dear author, 

Thanks for submitting your manuscript. See below for my comments.

Overall comments:

1: It is quite hard to claim an effect on such a multifactorial problem. There is no causal cause between GIS and specific gut damage markers.

Thank you for your comment. While we agree that endothelial disruption does not directly cause gastrointestinal symptoms, the proposed sequence of physiological changes laid out in by Van Wijck (2012) and then Costa et al (2017) in Exercise Associated Gastrointestinal syndrome does link the initial endothelial damage to changes in endotoxin levels (Costa et al., 2017; Wijck, van et al., 2012). Endotoxin levels are not consistently associated to gastrointestinal symptoms, but may be part of the cumulative pathology and still bares investigating. We have added some further discussion around this to the introduction (L77-94), which we hope further clarifies these points. 

2: Methods of included studies can be described more in depth to increase clarity.

Thank you for highlighting this. We have aimed to include as many aspects as possible of the various methodology in table 1, including exercise and supplementation protocol, participant details, as well as the data derived from the key biochemical markers. In order to ensure readability of the manuscript, this information is not repeated in text in order to limit the length of the paper. 

3: With high carbohydrate intakes we often see fructose-intolerances (or even fructose allergics) which also leads to GIS: how are you sure about excluding these individuals?

Thank you for highlighting this. Due to each study’s own selection criteria, the participants are included as healthy and without any pre-existing GI disorders. This may assist in selecting out those who may suffer from a fructose intolerance or known reaction during exercise. 

It is worth discussing that the addition of fructose to an existing glucose or maltodextrin drink may allow for higher absolute carbohydrate absorption and lower residual carbohydrate in the large colon, creating less risk for GIS (Oliveira, de & Burini, 2014). This seems to agree with some of the data from Wilson (2015) where rates of GIS were likely lower in the runners consuming a glucose/fructose mix compared to glucose alone (Wilson & Ingraham, 2015). The study by Flood et al (2020) is the only study in the review known by the authors to include a maltodextrin/fructose mix that exceeds ~60g/hour, with the other studies generally being lower in concentration or volume (Sessions et al., 2016). This may results in the other studies being less likely to have malabsorption or fructose-related GIS. Lastly, when using trained participants, there is a possibility that the small intestine upregulates the fructose and glucose transport proteins, again reinforcing that there is less risk of fructose being the direct cause for an increase in GIS (Horowitz et al., 1996). Lastly, we aimed to be clear that we were not evaluating the impact of supplements on GIS, and that fructose as a saccharide is unlikely to react or have different impact compared to other saccharides at the endothelial level in the small intestine. 

Specific comments: 

L77-89: The pathophysiology is poorly described. Please take a look at papers like de Oliveira et al 2014 and Costa et al 2016 and rewrite this specific part.

Thank you for your comment. We have adjusted the explanation; it now reads: 

Line 77: “(Wijck, van et al., 2011). Endothelial cell damage increases permeability via disruption and dysregulation of tight junction proteins (Davison et al., 2016; Dokladny et al., 2016). . Increased permeability has been associated with changes in urinary metabolites (Phua et al., 2015) and inflammatory markers (Karl et al., 2017).”

L158-159: Is the DSAT validated? If so, what are the pro and cons of this method?

The DSAT has been validated against isotopes method (Cr- EDTA/I4C-mannitol) and delayed-release, methacrylate-coated capsules in healthy non-exercises as a measure of gastrointestinal permeability (Blomquist et al., 1993; Camilleri et al., 2011) and more recently by Ogden and colleagues in healthy exercising men (Ogden et al., 2020). These references were included to illustrate the previous work in the DSAT validation. 

Although there is no standardised protocol and the length of collection time for urine are identified as challenges in athletes, urinary appearance of non-digestible saccharides is the most useful test to date to evaluate gastrointestinal cell permeability. These limitations have been highlighted in the end of the review. 

L224-225: i-FABP post exercise is not a reliable marker. You need te pre exercise value to get insights in the increase of i-FABP. Please come back to this in the discussion part.

Thank you for highlighting this; we have corrected the sentence as we were referring to the change from baseline rather than just the post-exercise value. The sentence now reads: 

Line 239: “Morrison et al (2014) showed a larger change from baseline to post-exercise(90mins) in i-FABP levels in trained compared to untrained participants, but showed no effect of colostrum, in spite of the high dosages given (>120g.day-1)” 

L260-263: What was in the CHO supplement? The effect of glucose for example is different that the effect of an olygosacharide.

Thank you for the comment. The details of each carbohydrate drink are in the overall summary table, but we have included this in the text as well for clarity. It now reads: 

Line 277: “Of note, Pugh and colleagues included exogenous carbohydrate supplementation during the exercise, due to the length of the exercise bout of ~66g·hr-1 (repeated 22g maltodextrin gels with 200ml water during the running marathon) and ~90g·hr-1 (10% maltodextrin fluid every 15 minutes during 120 minutes of cycling) in the four-strain probiotic studies (Pugh et al., 2019; Pugh et al., 2020).”

L278: there are different kind of measurements taken (blood i-FABP and urinary-i-FABP) which makes the results very questionable.

We agree, but felt that it was premature to exclude the paper. We have highlighted in the discussion that this is not considered a validated measure for endothelial damage. In the discussion, we have added to our previous comment to add clarity: 

Line 532: “While the study showed a benefit in lower i-FABP levels post 3000m time trial (~9-10 minutes) compared to standard protein supplementation, the study used urinary i-FABP as the marker of endothelial damage adjusted for urinary creatine levels. This may highlight renal clearance of i-FABP rather than systemic appearance, and may require prudent interpretation. This measure has yet to be validated elsewhere and may contribute to the >10 fold increase in concentration.”

L305-311: Rewrite this please. However the risk of bias is low, it has to be undelined that the outcomes are based on specific circumstances.

Thank you for highlighting this, we have added further detail to improve the details around the risk of bias analysis: 

Line 327: “Overall, the risk of bias across the selected studies was considered low. This was especially clear for sequence generation, allocation concealment and blinding of participants. Four of the twenty-seven studies (15%) used placebo-controlled or matched pairs designs (no crossover), while all other studies used randomised blinded crossover protocols. Due to the nature of the trials, blinding of outcomes was unclear. Five of the studies were registered as clinical trials, meaning that the risk of selective reporting for the remainder was unclear. The risk of bias summary for all studies can be found in Figure 2, and the risk of bias for individual studies are presented in the supplementary data.”

L317-319: Please dig deeper upon the saccharide types.

Thank you for your comment. Unfortunately, the studies that concern mono or disaccharide supplementation are so varied that they allow for very limited comparison. There is no evidence that saccharides will be treated differently at the level of the endothelial past the specific transporters. 

L383-384: I would like to see a discourage of usage because of the mentioned reasons.

The authors did not feel it was within our role to make any such statement on this topic because not all athletes are bound by the WADA anti-doping guidelines. We have presented the recent research and have clearly stated the current WADA warnings as objectively as possible, which we believe is appropriate to signpost athlete and allow them to make informed judgements as appropriate. 

L409-410: Why is only usage in high temperature highlighted? There are multiple practical implications you could think of.

Our identification of the heated environment is based on the included studies, where all four used heated conditions. While there may be advantages at thermoneutral conditions, none of the studies can confirm this. While there may be other research around glutamine and its use in colonocyte health, these studies are generally not evaluating endothelial dysfunction. 

L482-490: What about fiber intake?

Fibre-rich foods would slow down gastric emptying, but may not have impact in relation to monosaccharide levels in the gut. There is no comparable research using food or fibre containing fluids in the acute phase endothelial response to exercise. This will be an interesting avenue to investigate but may contradict the other performance nutrition research and requirements around exercise. We have added a note: 

Line 521: “Many of the studies target a low FODMAP pre-exercise meal, but there is also no understanding as yet of fibre (in solid or fluid) intake prior to exercise and any impact of gastric emptying on endothelial cell support.”

L614-618: the conclusion is missing the specific environmental settings. Please expend 'context dependent' with some of the factors that may influence GIS.

Thank you for your comment. We have added further details: 

Line 649: “As such, many of the supplements presented here should be evaluated for their effectiveness in a context dependent manner. External factors such as previous GIS, exercise modality, duration and intra-event fuelling opportunities should contribute to decision making.”

Reviewer #2: The authors provide a systematic review of a variety of dietary supplements and their impact on exercise-associated gut damage and gut permeability, primarily measured via Intestinal fatty acid binding protein (i-FABP) and/or dual saccharide absorption tests. The topic is of interested and the review methodology is strong following the PRISMA guidelines. A few suggestions are provided by section below. 

Thank you for your time and suggestions to our review. 

Introduction

Line 86- It would be nice to have more clarity regarding specifically, which gastrointestinal symptoms are associated with increased endotoxin levels, as well as, an estimated time to on-set and duration of the effects.

Line 86: Endotoxin levels have been associated with nausea in ultra-endurance athletes (Stuempfle et al., 2016) but not in shorter duration exercise (Moncada-Jiménez et al., 2009). As a consequence, the local and systemic responses may contribute to higher rates of gastrointestinal symptoms (Stuempfle et al., 2016).

Line 88 – Changes in gastric emptying and motility may affect nutrient absorption. Consider discussing, as this may affect gut permeability and be relevant to the saccharide absorption tests.

Thank you for highlighting this. We have added a sentence as such to expand on this concept within the research

Line 91: “Gastroparesis or ‘slosh stomach’ may contribute to reduced nutrient absorption or gastrointestinal cell injury, but the overall rates and consequences in athletes are unknown (Biondich & Joslin, 2016).”

Line 99 – Given that some of the studies use an acute supplementation and chronic, it would be valuable to discuss potential mechanism and/ore effects of both. For instance how would an chronic supplementation impact gut permeability during a bout of exercise vs. an acute protocol.

This is an interesting comment and we have added a sentence to add to the introduction. 

Line 104: “Further, the impact of chronic or acute supplementation protocols with similar supplements has not been considered.”

Methods

Why were your outcome measures not included in your “key word” search list?

Thank you for your comment. Due to the original scope of the review (to be as wide as possible), we made allowances for other markers to be used. This included fecal zonulin, fecal Calprotectin, or Iohexol. However, in the review process, we discovered that the DSAT and i-FABP tests were the most appropriate options to approximate endothelial damage and permeability, and that other markers were not used in athlete populations. 

Line 138 – It would help the reader to clarify that the ratio of lactulose to rhamnose/ mannose is used and that an increase in lactulose indicates increased permeability.

Thank you for highlighting. We have added some further explanation. It now reads: 

Line 144 “Lactulose (disaccharide) absorption and appearance in the urine/plasma will increase with disturbances in endothelial gap junctions while mannitol/rhamnose (monosaccharide) absorption is used as a normalising factor for lactulose since it utilises paracellular transport. An increase in the ratio of the larger disaccharide to the smaller monosaccharide illustrates and increase in paracellular passage and hence intestinal permeability.”

Line 164 – inclusion criteria for the exercise would be valuable if there were any. i.e. endurance exercise etc. It appears there are no studies looking at strength or intermittent exercise. Note if they were specifically excluded or simply not available. I see there is some discussion on this in the limitations with respect to “team” sports. 

This is a good observation since there are currently no studies evaluating these markers in team sports or resistance exercise. The single study in resistance exercise (Wijck, van et al., 2013) was excluded due to the lack of control group with regards to the whey protein intake. The study showed an increase in permeability and endothelial damage presented as part of the meta-analysis by the authors (Chantler et al., 2020). This is a large gap in the current research and one aspect of our future research. 

Finally, were there any criteria regarding the amount of exercise pre-test day i.e. after a “rest day” or dietary controls i.e. “no caffeine” or drug controls i.e. “no NSAIDS”?

While this is standardised amongst the research studies, we have included for clarity: 

Line 150: “To improve the robustness of these markers, participants are asked to avoid alcohol, spicy food, non-steroidal anti-inflammatories, caffeine or high intensity exercise in the day preceding exercise trials (Costa et al., 2017; Ogden et al., 2020).”

Results

Table 1 is comprehensive and well done. Table description for clarity confirm that difference is significant and p-value used.

Thank you, we have added this to the legend of the table. 

Table 1 – last row/column should “an” be “and”?

Thank you for spotting this – it has been corrected

Line 281 - Macronutrients can any statements be made regarding liquid vs solid based on the data?

The exercise and carbohydrate protocols are diverse and too limited in number to be able to make any comment with reasonable assertion. We have added more detail to this paragraph to clarify: 

Line 287: “Four studies using carbohydrate fluids (of various concentrations; 6-16% solution) mitigated the impact of exercise on i-FABP concentrations (Flood et al., 2020; Jonvik et al., 2019; Lambert et al., 2008; Snipe et al., 2017) although all used different saccharide combinations and concentrations limiting comparison (specific values in Table 1).”

Discussion

Bovine Colostrum – some insight into potential mechanism would be of interest if available. The in vitro studies do mention potential protective measures, but how these could be occurring is not discussed.

Thank you for your comment. We have briefly mentioned some of the mechanistic data (Line 354) and due to the overall length of the review felt it was important to limit the length of these explanations. 

We have added one additional sentence to reference further reviews of the research in this area: 

Line 362: “Some of the mechanisms have been investigated and reviewed (58) but will require further research around exercise to elucidate further.”

Line 356 – “affect” should be “effect”

It is odd that in line 305 it was noted that subgroup analysis could not occur and yet later it is noted that training status did not alter the response (line 368). I’m not sure there is sufficient data here to support the statement.

Thank you for the comment. The comparison of trained and untrained groups was derived from the study itself, not from any subgroup analysis by the authors. It is highlighted later in the paragraph that this is the only study to consider this question so far, and that further research would be useful to elucidate details around the role of fitness or training and gastrointestinal splanchnic response and consequent GI damage and permeability. We have adjusted the paragraph for clarity: 

Line 390: “In the single study that considered trained and untrained participants, training status did not alter the response to bovine colostrum supplementation compared to the placebo condition. The authors did note a higher i-FABP concentration post-exercise in the trained participants compared to untrained, but this was independent of colostrum supplementation (Morrison et al., 2014). Higher output during exercise based on training status reflected in body temperature could have contributed to this difference, but there was no significant difference in the body temperature during exercise between groups.”

Line 383 “it’s” should be “its”

Thank you, this has been changed. 

Consider expanding the discussion on heat shock protein and occludin to relate back to the current studies. One or two sentences to explain to the reader how these affect gut permeability should be sufficient.

Thank you for your comment. We have added an additional sentence to link the mechanisms to the possible benefits. 

Line 435: “Heat shock protein upregulation can improve cell tolerance to oxidative stress and inflammation and interact with occludin to preserve tight junction integrity (Arnal & Lallés, 2016). This is reflected in the lower permeability markers (L:R) levels seen by Zuhl and colleagues (Zuhl, M. et al., 2015; Zuhl, M. N. et al., 2014).”

Line 574 – “marker” should be “markers”

Thank you for spotting this error. It has been changed

Limitations

Potentially lack of controls in the study regarding caffeine and NSAIDs as mentioned in methods sections.

This has been added to the methods to clarify as it is generally standard practice in the included studies. 

Line 150: “To improve the robustness of these markers, it is standard practice for participants to avoid alcohol, spicy food, non-steroidal anti-inflammatories, caffeine or high intensity exercise in the day preceding exercise trials (Costa et al., 2017; Ogden et al., 2020).”

Line 597 – I also didn’t see studies in strength based or HITT/cross-fit type workouts

Please see our previous comment – since there are no studies to date to the authors knowledge. 

Line 601 – females generally experience higher levels of exercise-induced GI symptoms than males.

While this is true in the earlier studies, the recent review by Pugh et al states that there may be confounding factors such as smaller representation in endurance events and lower training age for women compared to men (Pugh et al., 2021). However, this is still worthwhile investigating further and is part of our future research. 

References: 

Arnal, M. E. & Lallés, J. P. (2016) Gut Epithelial Inducible Heat-Shock Proteins and Their Modulation by Diet and the Microbiota. Nutrition Reviews, 74 (3), pp. 181–197.

Biondich, A. S. & Joslin, J. D. (2016) Quantifying the “Slosh Stomach”: A Novel Tool for Assessment of Exercise-Associated Gastroparesis Symptoms in Endurance Athletes. Journal of Sports Medicine, 2016, pp. 1–5.

Blomquist, L., Bark, T., Hedenborg, G., Svenberg, T. & Norman, A. (1993) Comparison between the Lactulose/Mannitol and 51cr-Ethylenediaminetetraacetic Acid/14c-Mannitol Methods for Intestinal Permeability Frequency Distribution Pattern and Variability of Markers and Marker Ratios in Healthy Subjects. Scandinavian Journal of Gastroenterology, 28 (3), pp. 274–280.

Camilleri, M., Nadeau, A., Lamsam, J., Linker Nord, S., Ryks, M., Burton, D., Sweetser, S., Zinsmeister, A. R. & Singh, R. J. (2011) Understanding Measurements of Intestinal Permeability in Healthy Humans with Urine Lactulose and Mannitol Excretion. Neurogastroenterol Motil, 22 (1), pp. 1–22.

Chantler, S., Griffiths, A., Matu, J., Davison, G., Jones, B. & Deighton, K. (2020) The Effects of Exercise on Indirect Markers of Gut Damage and Permeability: A Systematic Review and Meta-Analysis. Sports Medicine.

Costa, R. J. S., Snipe, R. M. J., Kitic, C. M. & Gibson, P. R. (2017) Systematic Review : Exercise-Induced Gastrointestinal Syndrome — Implications for Health and Intestinal Disease. Alimentary pharmacology & therapeutics, (March), pp. 246–265.

Davison, G., Marchbank, T., March, D. S., Thatcher, R. & Playford, R. J. (2016) Zinc Carnosine Works with Bovine Colostrum in Truncating Heavy Exercise-Induced Increase in Gut Permeability in Healthy Volunteers. American Journal of Clinical Nutrition, 104 (2) August, pp. 526–536.

Dokladny, K., Zuhl, M. N. & Moseley, P. L. (2016) Intestinal Epithelial Barrier Function and Tight Junction Proteins with Heat and Exercise. Journal of Applied Physiology, 120 (6) March, pp. 692–701.

Flood, T. R., Montanari, S., Wicks, M., Blanchard, J., Sharpe, H., Taylor, L., Kuennen, M. R. & Lee, B. J. (2020) Addition of Pectin-Alginate to a Carbohydrate Beverage Does Not Maintain Gastrointestinal Barrier Function during Exercise in Hot-Humid Conditions Better than Carbohydrate Ingestion Alone. Applied Physiology, Nutrition, and Metabolism, pp. 1–37.

Horowitz, M., Cunningham, K. M., Wishart, J. M., Jones, K. L. & Read, N. W. (1996) The Effect of Short-Term Dietary Supplementation with Glucose on Gastric Emptying of Glucose and Fructose and Oral Glucose Tolerance in Normal Subjects. Diabetologia, 39 (4), pp. 481–486.

Jonvik, K. L., Lenaerts, K., Smeets, J. S. J., Kolkman, J. J., Loon, L. J. C. Van & Verdijk, L. B. (2019) Sucrose but Not Nitrate Ingestion Reduces Strenuous Cycling-Induced Intestinal Injury. Medicine and Science in Sports and Exercise, 51 (3), pp. 436–444.

Karl, J. P., Margolis, L. M., Madslien, E. H., Murphy, N. E., Castellani, J. W., Gundersen, Y., Hoke, A. V., Levangie, M. W., Kumar, R., Chakraborty, N., Gautam, A., Hammamieh, R., Martini, S., Montain, S. J. & Pasiakos, S. M. (2017) Changes in Intestinal Microbiota Composition and Metabolism Coincide with Increased Intestinal Permeability in Young Adults under Prolonged Physiological Stress. American Journal of Physiology - Gastrointestinal and Liver Physiology, 312 (6) June, pp. G559–G571.

Lambert, G. P., Lang, J., Bull, A., Pfeifer, P. C., Eckerson, J., Moore, C., Lanspa, S. & O’Brien, J. (2008) Fluid Restriction during Running Increases GI Permeability. International Journal of Sports Medicine, 29 (3) March, pp. 194–198.

Moncada-Jiménez, J., Plaisance, E., Mestek, M. L., Araya-Ramírez, F., Ratcliff, L., Taylor, J. K., Grandjean, P. W. & AragónVargas, L. F. (2009) Initial Metabolic State and Exercise-Induced Endotoxaemia Are Unrelated to Gastrointestinal Symptoms during Exercise. Journal of Sports Science and Medicine, 8 (2) June, pp. 252–258.

Morrison, S. A., Cheung, S. S. & Cotter, J. D. (2014) Bovine Colostrum, Training Status, and Gastrointestinal Permeability during Exercise in the Heat: A Placebo-Controlled Double-Blind Study. Applied Physiology, Nutrition and Metabolism, 39 (9), pp. 1070–1082.

Ogden, H. B., Fallowfield, J. L., Child, R. B., Davison, G., Fleming, S. C., Edinburgh, R. M., Delves, S. K., Millyard, A., Westwood, C. S. & Layden, J. D. (2020) Reliability of Gastrointestinal Barrier Integrity and Microbial Translocation Biomarkers at Rest and Following Exertional Heat Stress. Physiological Reports, 8 (5), pp. 1–16.

Oliveira, E. P. de & Burini, R. C. (2014) Carbohydrate-Dependent, Exercise-Induced Gastrointestinal Distress. Nutrients, 6 (10), pp. 4191–4199.

Phua, L. C., Wilder-Smith, C. H., Tan, Y. M., Gopalakrishnan, T., Wong, R. K., Li, X., Kan, M. E., Lu, J., Keshavarzian, A. & Chan, E. C. Y. (2015) Gastrointestinal Symptoms and Altered Intestinal Permeability Induced by Combat Training Are Associated with Distinct Metabotypic Changes. Journal of Proteome Research, 14 (11) November, pp. 4734–4742.

Pugh, J. N., Lydon, K., O’Donovan, C. M., O’Sullivan, O. & Madigan, S. M. (2021) More than a Gut Feeling: What Is the Role of the Gastrointestinal Tract in Female Athlete Health? European Journal of Sport Science, pp. 1–10.

Pugh, J. N., Sparks, A. S., Doran, D. A., Fleming, S. C., Langan-Evans, C., Kirk, B., Fearn, R., Morton, J. P. & Close, G. L. (2019) Four Weeks of Probiotic Supplementation Reduces GI Symptoms during a Marathon Race. European Journal of Applied Physiology, 119 (7), pp. 1491–1501.

Pugh, J. N., Wagenmakers, A. J. M., Doran, D. A., Fleming, S. C., Fielding, B. A., Morton, J. P. & Close, G. L. (2020) Probiotic Supplementation Increases Carbohydrate Metabolism in Trained Male Cyclists: A Randomized, Double-Blind, Placebo-Controlled Crossover Trial. American Journal of Physiology - Endocrinology and Metabolism, 318 (4), pp. E504–E513.

Sessions, J., Bourbeau, K., Rosinski, M., Szczygiel, T., Nelson, R., Sharma, N. & Zuhl, M. (2016) Carbohydrate Gel Ingestion during Running in the Heat on Markers of Gastrointestinal Distress. European Journal of Sport Science, 16 (8) November, pp. 1064–1072.

Snipe, R. M. J., Khoo, A., Kitic, C. M., Gibson, P. R. & Costa, R. J. S. (2017) Carbohydrate and Protein Intake during Exertional-Heat Stress Ameliorates Intestinal Epithelial Injury and Small Intestine Permeability. Appl. Physiol. Nutr. Metab, 42 (November) August, pp. 1–41.

Stuempfle, K. J., Valentino, T., Hew-Butler, T., Hecht, F. M. & Hoffman, M. D. (2016) Nausea Is Associated with Endotoxemia during a 161-Km Ultramarathon. Journal of Sports Sciences, 34 (17), pp. 1662–1668.

Wijck, K. van, Lenaerts, K., Grootjans, J., Wijnands, K. A. P., Poeze, M., Loon, L. J. C. van, Dejong, C. H. C. & Buurman, W. A. (2012) Physiology and Pathophysiology of Splanchnic Hypoperfusion and Intestinal Injury during Exercise: Strategies for Evaluation and Prevention. American Journal of Physiology - Gastrointestinal and Liver Physiology, 303 (2) July, pp. G155–G168.

Wijck, K. van, Lenaerts, K., Loon, L. J. C. van, Peters, W. H. M., Buurman, W. A. & Dejong, C. H. C. (2011) Exercise-Induced Splanchnic Hypoperfusion Results in Gut Dysfunction in Healthy Men. PLoS ONE, 6 (7) July, p. e22366.

Wijck, K. van, Pennings, B., Bijnen, A. A. van, Senden, J. M. G. G., Buurman, W. A., Dejong, C. H. C. C., Loon, L. J. C. van, Lenaerts, K., Wijck, K. Van, Pennings, B., Bijnen, A. A. Van, Senden, J. M. G. G., Buurman, W. A., Dejong, C. H. C. C., Loon, L. J. C. Van, Lenaerts, K., Wijck, K. van, Pennings, B., Bijnen, A. A. van, Senden, J. M. G. G., Buurman, W. A., Dejong, C. H. C. C., Loon, L. J. C. van & Lenaerts, K. (2013) Dietary Protein Digestion and Absorption Are Impaired during Acute Postexercise Recovery in Young Men. American Journal of Physiology - Regulatory Integrative and Comparative Physiology, 304 (5), pp. 356–361.

Wilson, P. B. & Ingraham, S. J. (2015) Glucose-Fructose Likely Improves Gastrointestinal Comfort and Endurance Running Performance Relative to Glucose-Only. Scandinavian Journal of Medicine and Science in Sports, 25 (6), pp. e613–e620.

Zuhl, M., Dokladny, K., Mermier, C., Schneider, S., Salgado, R. & Moseley, P. (2015) The Effects of Acute Oral Glutamine Supplementation on Exercise-Induced Gastrointestinal Permeability and Heat Shock Protein Expression in Peripheral Blood Mononuclear Cells. Cell Stress and Chaperones, 20 (1), pp. 85–93.

Zuhl, M. N., Lanphere, K. R., Kravitz, L., Mermier, C. M., Schneider, S., Dokladny, K. & Moseley, P. L. (2014) Effects of Oral Glutamine Supplementation on Exercise-Induced Gastrointestinal Permeability and Tight Junction Protein Expression. Journal of Applied Physiology, 116 (2) January, pp. 183–191.

---

## [Decision Letter · Decision Letter 1]

23 Nov 2021

PONE-D-21-15349R1A SYSTEMATIC REVIEW: ROLE OF DIETARY SUPPLEMENTS ON MARKERS OF EXERCISE-ASSOCIATED GUT DAMAGE AND PERMEABILITYPLOS ONE

Dear Dr. Chantler,

Thank you for resubmitting your manuscript to PLOS ONE and making your manuscript significantly improved.

After careful consideration, we feel that it has merit but does not fully meet PLOS ONE’s publication criteria as it currently stands. The work requires additional compliance with the recommendations of the reviewer and the editor's comments.

Therefore, we invite you to submit a revised version of the manuscript that addresses the points raised during the review process.

We look forward to receiving your revised manuscript.

Kind regards,

Krzysztof Durkalec-Michalski, Ph.D

Academic Editor

PLOS ONE

Journal Requirements:

Additional Editor Comments (Please be aware that comments are referenced to text with the track changes option):

Line 41 – it should be clearly indicated that authors mention reduction in “selected” markers.Conclusion, taking into account the large heterogeneous group of treatments (various supplements, euhydration, carbohydrates) should be formulated much more carefully.Line 66 and 128 - the consent number of the indicated bioethics committee is missing.Line 87 – the double space should be removed.Line 89 - The double dot must be removed.Lines 248, 249 and 251 - standardize spacing (insert space).In tables, other co-authors should be respected and where applicable, "et al." should be written next to the first author's name.Line 91 - as you indicate that it is mean values - remove the unnecessary standard deviation.Line 111 - It should be pointed out that "nutritional support" rather than only "supplementation" is particularly important in the discussed topic.Line 138 – the authors wrote that “database can be found in supplementary material” – Please indicate clearly where exactly these data are.Line 152 – “an increase in plasma concentration” – but “an increase” of what? Please clarify.Please insert "i-FABP" for the first time when using "Intestinal fatty acid binding protein" for the first time.Line 165 – the authors wrote: “Supplements, by the IOC definition, can include any food” - Food is not the same as supplements. These concepts should not be confused.Line 160 – the space between “/” and “low” should be removed.Lines 182-185 - This section is convoluted and worth refuting. Besides, the authors write about "a number of studies" and cite one work.Line e.g. 182 and where applicable - Please make sure that "dual saccharides" is the correct statement in this context and that it should not be "disaccharides".Line 197 – the authors wrote: “In total, thirty studies met the inclusion criteria”. However, in the pointed figure 1 or in the line 230,  27 works are indicated. Please correct properly throughout the work truthfully.Lines 205-208 – second part of this sentence is quite difficult to follow. Please consider to clarity this.Lines 225-227 – the authors wrote "Based on previous work ..." - however, there are no citations in this regard. This should be completed.Lines 246-248 - The sentence is a bit vague - please clarify. it is a bit incomprehensible what this temperature is about - is it exercises at this temperature?The notation of units is not uniform at work, e.g. in line 251 the authors write "> 120g.day^-1^) (39)" and in line 246 "(20g · day^-1^") - it should be standardized (here it is about "dot").The tables require very thorough corrections. They are not uniform when it comes to editing and description. In some sections no clear marker is indicated, intervals are not uniform, no indication of what was given as placebo, gaps of necessary units for some markers, spacing, missing or incomplete parentheses, no indication of "et al." after the first authors, it is not clear in some records whether powder or capsules were administered, the significance value is missing in some records, some units due to strange notation unclear, the accuracies of the reported data within some records should also be standardized. Furthermore, strains of probiotics should be clearly stated.On page 12 (Table 1) of Pugh et al. (2017), the authors misinterpreted the amount of glutamine (09g…). Do I understand correctly that it should be 0.9 g…?On page 12 (Table 1) - unit for correction in Osborne et al. (2019). Now it is: "ƞg·ml^-1^".Unclear sign (page 13, table 1, Mooren et al. (2020): "<" before the age of the respondents ("18-35 years").What does "placebo diet" mean? (page 14, table 1). In addition, in this record also probably lacks the results of one group (Karl et al., (2017).With some records, the work number (from the references) is not indicated (e.g. with "Ma et al. (2020)").Additionally, in the "Ma et al. (2020)" record, there is an error in the writing of markers - writing and using "/" suggests that it is "i-FABP / Creatine ratio". Besides, the authors spelled it wrong - there shouldn't be "Creatine" but "Creatinine". The above example shows that all markers in the table and the work should be carefully checked by the Authors for serious mistakes in this respect.Table 1, page 15 (Kartaram et al. (2018) and line 327 - what do the authors mean "ad libertum"? Shouldn't it be "ad libitum"?Line 272 – correct dose - shouldn't it be like in the table (0.25g…)?Line 274 - why "likely" is in italics?Line 279 – correct unit.Line 402 - some blue color is inserted.Line 407 and 412 – remove the double dot.Lines 414-419 - The topic of Colostrum and doping issues and WADA is a repeated false myth. You should clearly state your factual position on this matter and not repeat these myths.The section on "Glutamine" - the work in a way recommends the effectiveness and use of GLU. However, the reader should clearly indicate the potential risk and the negative one, especially related to the supply of high doses. It has been written extensively about this in the literature.Line 445 – change “…by Zuhl and colleagues (43,44) This may highlight” to “…by Zuhl and colleagues (43,44), and may highlight”.Line 489 – use the impersonal form in this sentence.Line 495 - insert the number of the cited work.Lines 515-516 - I do not agree with the authors on this point, because there has been some work in this regard - they should be reached.Line 546 - "creatine" or "creatinine" levels?Line 563 – correct the abbreviation.In the conclusions you generally write about some specific "carbohydrate" or should be "carbohydrates"?Notation of some references should be corrected.The surname "de Oliveira" should be spelled "de Oliveira" not "Oliveira de". Should be corrected in the literature record and where applicable.

Reviewers' comments:

Reviewer's Responses to Questions

**Comments to the Author**

1. If the authors have adequately addressed your comments raised in a previous round of review and you feel that this manuscript is now acceptable for publication, you may indicate that here to bypass the “Comments to the Author” section, enter your conflict of interest statement in the “Confidential to Editor” section, and submit your "Accept" recommendation.

Reviewer #1: All comments have been addressed

Reviewer #2: All comments have been addressed

Reviewer #3: (No Response)

2. Is the manuscript technically sound, and do the data support the conclusions?

Reviewer #1: Yes

Reviewer #2: Yes

Reviewer #3: Yes

3. Has the statistical analysis been performed appropriately and rigorously? 

Reviewer #1: Yes

Reviewer #2: N/A

Reviewer #3: N/A

4. Have the authors made all data underlying the findings in their manuscript fully available?

Reviewer #1: Yes

Reviewer #2: Yes

Reviewer #3: Yes

5. Is the manuscript presented in an intelligible fashion and written in standard English?

Reviewer #1: Yes

Reviewer #2: Yes

Reviewer #3: Yes

6. Review Comments to the Author

Reviewer #1: (No Response)

Reviewer #2: (No Response)

Reviewer #3: Overall:

Reviewer #2: The authors provide a systematic review of a variety of dietary supplements and their impact on exercise-associated gut damage and gut permeability. The topic is highly interesting. A few suggestions are provided by section below.

Nevertheless, all the chosen supplements were just briefly explained in this paper. The results are mainly in the table. There are no characteristics of the chosen substances, nor the mechanisms underlying their effects on gastrointestinal problems during exercise, nor the explanation why these substances where chosen for this review (except the fact that there are studies using this supplements for GI tract problems). In my opinion there are too many, not related to each other (for example fluid restrictions in the same review as glutamine), kinds of supplements investigated just briefly in this review.

Introduction

Line 95-97: You mention just the increased immune response, not a state called immunodepression, which is even a more significant problem after a bout of acute exercise.

Line 97-98: What does a shorter duration exercise mean? In comparison to previously mentioned ultra-endurance and long distance.

Methods

Line 132: I can see what keywords were used to perform the search, but nothing is mentioned about the exact index terms that were used.

165-177: I don’t quite understand why this paragraph is in the Methods section rather than in the Introduction.

Line 181: Was the type of placebo used in the research important for this Review outcomes?

Line 188-189: Why there were no restrictions placed on the training status of participants?

Results:

Line 234-235: “Nineteen studies (70%) of the studies used..” – there is something wrong with the style.

Line 246: You didn’t mention whether bovine colostrum dosages where acute or chronic.

Line 251: Why did you mention that the dosage in this study was > 120 g/day? It was 1,7 g/kg/day and the average body mass was 75 kg in trained participants.

What is more important about the colostrum supplementation studies is the placebo that is chosen. You didn’t mention it at all. It can make a huge difference when analysing the results of the supplementation vs placebo.

332-341: Why is this paragraph placed right after the results of hydration status? It is quite odd for me and makes the reading of this review difficult.

Discussion

361-363: I would expect thorough explanation of the mechanisms of colostrum on the gastrointestinal functions, not only mentioning and briefly describing that there are in vitro and in vivo studies.

Again there is nothing mentioned about the placebo used in the studies with colostrum supplementation.

408-409: “therefore, it is unknown if training status alters the gut perfusion-exercise relationship” – are you confident enough to put this statement?

410: Are you sure about the dosage?

414-419: In my opinion, based on the previous research by Jones et al. you should reconsider putting this statement in this review, because it mas be understood that colostrum is a banned substance that shouldn’t be supplemented, which is completely untrue.

437-438 and 513-514: There is suddenly an animal based study mentioned, again just briefly, and not in the context of every supplement you mentioned in this Review.

458: I am convinced that nowadays we don’t use the term bacterial flora, instead you should write microbiota, in my opinion.

7. PLOS authors have the option to publish the peer review history of their article (what does this mean?). If published, this will include your full peer review and any attached files.

Reviewer #1: **Yes: **Daan Hoogervorst

Reviewer #2: **Yes: **Jill Parnell

Reviewer #3: No

---

## [Author Response · Author response to Decision Letter 1]

22 Dec 2021

Dear Krzysztof Durkalec-Michalski and editorial team, 

RE: Response to reviewers

Thank you for giving us the opportunity to revise our manuscript again. We would like to thank the reviewers for their suggestions and comments, which we feel have enhanced the quality and accuracy of the manuscript. 

We have responded to all the comments and suggestions individually below. All modifications in the manuscript have been made using track-changes as requested. For clarity, the page numbers referred to in this reply are based on the track changes copy, and not the clean copy. 

Thank you again for your time and assistance. 

Best regards, 

Sarah Chantler

On behalf of the authors 

 

Response to the reviewers: 

Thank you for taking the time to review the manuscript again. We have aimed to address your concerns below and hope you will find our changes to your satisfaction. 

Editor Comments

Thank you for your input. I have added ‘amended’ to the comments that were changed without any need for further discussion. 

1. Line 41 – it should be clearly indicated that authors mention reduction in “selected” markers. Amended

2. Conclusion, taking into account the large heterogeneous group of treatments (various supplements, euhydration, carbohydrates) should be formulated much more carefully.

Thank you, the conclusion of the abstract has been adjusted accordingly to reflect your comment: 

Line 47: “In spite of a large heterogeneity across the selected studies, appropriate inclusion of different nutrition strategies could mitigate the initial phases of gastrointestinal cell disturbances in athletes associated with exercise.”

3. Line 66 and 128 - the consent number of the indicated bioethics committee is missing. Added

4. Line 87 – the double space should be removed. Amended

5. Line 89 - The double dot must be removed. Amended

6. Lines 248, 249 and 251 - standardize spacing (insert space). Amended

7. In tables, other co-authors should be respected and where applicable, "et al." should be written next to the first author's name. Amended

8. Line 91 - as you indicate that it is mean values - remove the unnecessary standard deviation. Amended

9. Line 111 - It should be pointed out that "nutritional support" rather than only "supplementation" is particularly important in the discussed topic. Amended

10. Line 138 – the authors wrote that “database can be found in supplementary material” – Please indicate clearly where exactly these data are.

Thank you, I have reformatted the indication for all supplementary materials as S1, S2, S3 as per the author guidelines. 

11. Line 152 – “an increase in plasma concentration” – but “an increase” of what? Please clarify. Clarified 

12. Please insert "i-FABP" for the first time when using "Intestinal fatty acid binding protein" for the first time. Amended

13. Line 165 – the authors wrote: “Supplements, by the IOC definition, can include any food” - Food is not the same as supplements. These concepts should not be confused.

This definition is directly attributed to the IOC position stand on supplementation. From the position stand: 

“…we recognise that dietary supplements come in many forms, including the following: 

1. functional foods, foods enriched with additional nutrients or components outside their typical nutrient composition (eg, mineral-fortified and vitamin-fortified, as well as nutrient- enriched foods)

2. formulated foods and sports foods, products providing energy and nutrients in a more convenient form than normal foods for general nutrition support (eg, liquid meal replacements) or for targeted use around exercise (eg, sports drinks, gels, bars)”

This definition therefore can include carbohydrate and protein, in both supplemental or sports food (carbohydrate gel) or food based form. Since a portion of the studies selected included nutrients in powder, capsule, sports food and food form – it was felt this definition would be appropriate (Karl et al, 2017). If the intention of the food or food derivative is to supplement the existing diet (e.g. around exercise) then it can be included in the definition. This gives clarity to the logic behind food changes (e.g. FODMAP or gluten) not being included in the review. 

14. Line 160 – the space between “/” and “low” should be removed. Amended

15. Lines 182-185 - This section is convoluted and worth refuting. Besides, the authors write about "a number of studies" and cite one work.

Thank you for your comment. The sentence has been rewritten to try improve clarity, with the other studies that did not include a resting control. It now reads: 

Line 191: “The studies that included DSAT in their methodology but had no resting control were excluded from data extraction for that marker (Flood et al., 2020; Pugh, J. N. et al., 2020; Snipe et al., 2017; Wijck, Van et al., 2014).”

16. Line e.g. 182 and where applicable - Please make sure that "dual saccharides" is the correct statement in this context and that it should not be "disaccharides". Thank you, I have written the full name of the test; dual saccharide absorption test. 

17. Line 197 – the authors wrote: “In total, thirty studies met the inclusion criteria”. However, in the pointed figure 1 or in the line 230, 27 works are indicated. Please correct properly throughout the work truthfully. Amended

18. Lines 205-208 – second part of this sentence is quite difficult to follow. Please consider to clarity this.

Thank you for highlighting this. The sentence has been changed to read: 

Line 206: “Data were extracted independently by two researchers (SC and AG) into a standardised spreadsheet, which included the characteristics of articles valid for review; including gastrointestinal damage and permeability markers.”

19. Lines 225-227 – the authors wrote "Based on previous work ..." - however, there are no citations in this regard. This should be completed. Amended

20. Lines 246-248 - The sentence is a bit vague - please clarify. it is a bit incomprehensible what this temperature is about - is it exercises at this temperature?

To improve the reading, the sentence now reads: 

Line 251: “Dosage-matched (20g·day-1) studies evaluating endothelial damage (i-FABP) showed a positive blunting effect of colostrum subsequent to exercising at 22 ˚C (March et al., 2017) and 30˚C (March et al., 2019) but not at 40˚C (McKenna et al., 2017).”

21. The notation of units is not uniform at work, e.g. in line 251 the authors write "> 120g.day-1) (39)" and in line 246 "(20g · day-1") - it should be standardized (here it is about "dot"). Changed

22. The tables require very thorough corrections. They are not uniform when it comes to editing and description. In some sections no clear marker is indicated, intervals are not uniform, no indication of what was given as placebo, gaps of necessary units for some markers, spacing, missing or incomplete parentheses, no indication of "et al." after the first authors, it is not clear in some records whether powder or capsules were administered, the significance value is missing in some records, some units due to strange notation unclear, the accuracies of the reported data within some records should also be standardized. Furthermore, strains of probiotics should be clearly stated.

Thank you for highlighting the errors and inconsistencies. We have made every effort to adjust the table to reflect the data extracted from the studies in the most consistent way possible. Units, spacing, authors (et al), placebos, significance values have been checked and aligned. 

23. On page 12 (Table 1) of Pugh et al. (2017), the authors misinterpreted the amount of glutamine (09g…). Do I understand correctly that it should be 0.9 g…? Amended

24. On page 12 (Table 1) - unit for correction in Osborne et al. (2019). Now it is: "ƞg·ml-1". Amended

25. Unclear sign (page 13, table 1, Mooren et al. (2020): "<" before the age of the respondents ("18-35 years"). Amended

26. What does "placebo diet" mean? (page 14, table 1). In addition, in this record also probably lacks the results of one group (Karl et al., (2017). 

Thank you. The table has been adjusted to reflect the placebo condition as the standard diet (14.6 MJ/~3500kcal·day) and to show the three groups in the findings more clearly. 

27. With some records, the work number (from the references) is not indicated (e.g. with "Ma et al. (2020)"). Amended

28. Additionally, in the "Ma et al. (2020)" record, there is an error in the writing of markers - writing and using "/" suggests that it is "i-FABP / Creatine ratio". Besides, the authors spelled it wrong - there shouldn't be "Creatine" but "Creatinine". The above example shows that all markers in the table and the work should be carefully checked by the Authors for serious mistakes in this respect. Thank you for highlighting this error. It has been amended.

29. Table 1, page 15 (Kartaram et al. (2018) and line 327 - what do the authors mean "ad libertum"? Shouldn't it be "ad libitum"? Amended

30. Line 272 – correct dose - shouldn't it be like in the table (0.25g…)? Amended

31. Line 274 - why "likely" is in italics? The paper uses magnitude-based inferences as their statistical model, and use likely in italics to differentiate from standard interpretation. To improve the understanding, it now reads: 

Line 280: “Magnitude based inferences suggested a likely lower i-FABP levels post-exercise in the 0.5 and 0.9g·kg-1 FFM conditions compared to the placebo (Pugh, J. et al., 2017).”

32. Line 279 – correct unit. Amended

33. Line 402 - some blue colour is inserted. Amended

34. Line 407 and 412 – remove the double dot. Amended

35. Lines 414-419 - The topic of Colostrum and doping issues and WADA is a repeated false myth. You should clearly state your factual position on this matter and not repeat these myths.

Thank you for your concern around this topic. The authors have aimed not to misrepresent any of the existing data on this topic. However, for practitioners working in sport; it is also important to acknowledge that this is still on the WADA website: https://www.wada-ama.org/en/questions-answers/prohibited-list-qa#item-388 Question 8. All risk management for athletes should have all the available information. We have removed the first sentence to allow for less possible confusion on this topic. 

36. The section on "Glutamine" - the work in a way recommends the effectiveness and use of GLU. However, the reader should clearly indicate the potential risk and the negative one, especially related to the supply of high doses. It has been written extensively about this in the literature.

Thank you for the suggestion. Recent literature has showed that dosages up to 0.9g·kg/day are tolerated at rest (Ogden et al., 2020) and there were no issues reported in the study by Pugh et al (2017) at the higher dosages. We have included an additional sentence to highlight this possible concern: 

Line 463: “Pugh et al (2017) illustrated a possible dose-dependent benefit but was the only study to compare different dosages. Higher dosages (0.9g·kg) had minimal impact on GIS in spite of some concerns around tolerance (Pugh, J. et al., 2017).“

We have included a further sentence around the justification on long term supplementation: 

Line 490: “To note, chronic high dosages are not recommended due to concerns over long term effects (Holecek, 2013).” 

37. Line 445 – change “…by Zuhl and colleagues (43,44) This may highlight” to “…by Zuhl and colleagues (43,44), and may highlight”. Amended

38. Line 489 – use the impersonal form in this sentence. Adjusted

39. Line 495 - insert the number of the cited work. Amended

40. Lines 515-516 - I do not agree with the authors on this point, because there has been some work in this regard - they should be reached. 

We are unaware of any research in humans that will illustrate the energy expenditure of the gastrointestinal lining in healthy participants, or around exercise and endothelial recovery. If the editor is aware of research in this area, we would greatly appreciate sharing. Due to concerns over inaccuracy, the text has been adjusted: 

Line 560:” These gastrointestinal associated energy costs may be higher in athletes over longer duration exercise.”

41. Line 546 - "creatine" or "creatinine" levels? Changed

42. Line 563 – correct the abbreviation. Amended

43. In the conclusions you generally write about some specific "carbohydrate" or should be "carbohydrates"? Amended

44. Notation of some references should be corrected. Amended

45. The surname "de Oliveira" should be spelled "de Oliveira" not "Oliveira de". Should be corrected in the literature record and where applicable. Apologies but I am unsure where this is in the reference list, as I have not referenced their work. 

Comments to the Author from Reviewer #3

Thank you for your comments and suggestions to improve the content of our paper. 

Reviewer #3: Overall:

Reviewer #2: The authors provide a systematic review of a variety of dietary supplements and their impact on exercise-associated gut damage and gut permeability. The topic is highly interesting. A few suggestions are provided by section below.

Nevertheless, all the chosen supplements were just briefly explained in this paper. The results are mainly in the table. There are no characteristics of the chosen substances, nor the mechanisms underlying their effects on gastrointestinal problems during exercise, nor the explanation why these substances where chosen for this review (except the fact that there are studies using this supplements for GI tract problems). In my opinion there are too many, not related to each other (for example fluid restrictions in the same review as glutamine), kinds of supplements investigated just briefly in this review.

Thank you for your comments. The authors agree that the review did cover multiple supplements, but that the choices are directed by the outcome measures. The choices were revealed by the search process rather than a more narrative review approach which could have been more selective around the mechanisms. 

Introduction

Line 95-97: You mention just the increased immune response, not a state called immunodepression, which is even a more significant problem after a bout of acute exercise.

Thank you for the suggested revision. We have a concern that immunodepression is a more general term associated with the changes post-exercise, often localised around the upper respiratory tract, and general inflammatory cytokines (Moreira et al., 2007). 

Due to the tenuous link between gastrointestinal cell damage and the expression of gastrointestinal symptoms, there is evidence for some immune response, but not necessarily immunodepression. We felt we did not want to over-estimate the immune response in this explanation, and therefore have not incorporated the suggested wording. 

Line 97-98: What does a shorter duration exercise mean? In comparison to previously mentioned ultra-endurance and long distance.

Thank you, the sentence has been adapted to clarify the timing – <2-3 hours which may be considered shorter than the ultra-endurance events. 

Methods

Line 132: I can see what keywords were used to perform the search, but nothing is mentioned about the exact index terms that were used. 

Thank you, these are in the search terms included in the supplementary material (S1). 

165-177: I don’t quite understand why this paragraph is in the Methods section rather than in the Introduction.

Thank you for your comment. When considering the variety of research available, it was important to differentiate between different inclusion and exclusion criteria. Therefore, the definition applied to the inclusion criteria would be more suitable in the methods rather than the introduction. 

Line 181: Was the type of placebo used in the research important for this Review outcomes?

Thank you for the question. We have included the content of the placebo substances into the table. When appropriate, the authors aimed to include this in the discussion around the limitations. 

Line 188-189: Why there were no restrictions placed on the training status of participants?

Thank you for the question. We have elaborated in Line 196: “No restrictions were placed on the training status or sex of the participants as current studies available do not indicate there are any sex-based differences or training effect (Morrison et al., 2014; Snipe & Costa, 2018). “

Morrison et al (2014) found no difference in the trained and untrained groups in response to supplementation; and Snipe et al found no significant differences in the response to exercise between the sexes. These limited studies do not give sufficient grounds to refine the inclusion criteria further. 

Results:

Line 234-235: “Nineteen studies (70%) of the studies used..” – there is something wrong with the style.

Thank you, this has been corrected. 

Line 246: You didn’t mention whether bovine colostrum dosages where acute or chronic.

Thank you. These details can be found in Table 1. We have included an additional sentence to assist with clarity: 

Line 256: “Studies ranged from 2 to 14 days.”

Further, there is a summary of the acute vs chronic protocols in the results that may be useful. 

Line 241: “Supplementation protocols ranged from a single dose pre and/or during exercise (n=12, 24 hours pre-exercise) to multiple days of daily supplementation (n=15; up to 8 weeks).”

Line 251: Why did you mention that the dosage in this study was > 120 g/day? It was 1,7 g/kg/day and the average body mass was 75 kg in trained participants.

We felt that due to the other studies all having a dosage in g·day-1 that it may be useful to understand the estimated comparison. As this may have been estimated incorrectly from the authors, we have removed it. 

What is more important about the colostrum supplementation studies is the placebo that is chosen. You didn’t mention it at all. It can make a huge difference when analysing the results of the supplementation vs placebo.

Thank you for highlighting this – we have included all the placebos used across the studies in Table 1. 

332-341: Why is this paragraph placed right after the results of hydration status? It is quite odd for me and makes the reading of this review difficult.

Thank you for your comment. We have moved the reporting of risk of bias earlier in the results to prevent any confusion. (Line 254)

Discussion

361-363: I would expect thorough explanation of the mechanisms of colostrum on the gastrointestinal functions, not only mentioning and briefly describing that there are in vitro and in vivo studies.

Again there is nothing mentioned about the placebo used in the studies with colostrum supplementation.

Thank you for your comment. We have aimed to reference other previous reviews where the mechanisms are covered in more detail. The authors aimed to balance the content provided on each supplement. However, we understand that Bovine Colostrum has a much more extensive literature base to work from. We felt that the variability in the colostrum content itself based on milking times post-partum would be contribute more to results that the specific placebo used. 

408-409: “therefore, it is unknown if training status alters the gut perfusion-exercise relationship” – are you confident enough to put this statement?

Thank you for your comment. Due to the single study, and limited research in this area, we have altered the wording to reflect something less categorical: 

Line 430: “All, but one study (Mooren et al., 2020), used trained participants, and therefore, there is limited understanding if training status alters the gut perfusion-exercise relationship.”

410: Are you sure about the dosage? 

Due to your concerns over the use of the mean weight reported in the study, we have removed this. 

414-419: In my opinion, based on the previous research by Jones et al. you should reconsider putting this statement in this review, because it mas be understood that colostrum is a banned substance that shouldn’t be supplemented, which is completely untrue.

Thank you for your opinion. We have removed the first sentence of the paragraph to limit any confusion on our position of the current status. However, the wording from WADA is still important:

The wording from WADA: https://www.wada-ama.org/en/questions-answers/prohibited-list-qa#item-388 : Colostrum is not specifically prohibited, however it contains certain quantities of IGF-1 and other growth factors which are prohibited and may influence the outcome of anti-doping tests. Therefore, WADA does not recommend the ingestion of this product.

For any practitioners or athletes, this is still to ensure that there is full transparency on the risk minimisation recommendations by WADA. 

437-438 and 513-514: There is suddenly an animal based study mentioned, again just briefly, and not in the context of every supplement you mentioned in this Review.

Thank you for your comment. We have taken your comment under advisement to limit any reference to animal studies where possible and selected only studies done in human subjects. 

458: I am convinced that nowadays we don’t use the term bacterial flora, instead you should write microbiota, in my opinion. Amended to reflect your suggestion.

---

## [Decision Letter · Decision Letter 2]

16 Feb 2022

PONE-D-21-15349R2A SYSTEMATIC REVIEW: ROLE OF DIETARY SUPPLEMENTS ON MARKERS OF EXERCISE-ASSOCIATED GUT DAMAGE AND PERMEABILITYPLOS ONE

Dear Dr.Chantler,

Thank you for submitting your manuscript to PLOS ONE. After careful consideration, we feel that revised manuscript has been significantly improved, but still needs some minor corrections (follow the editor's notes). Therefore, we invite you to submit a revised version of the manuscript that addresses the points raised during the review process.

We look forward to receiving your revised manuscript.

Kind regards,

Krzysztof Durkalec-Michalski, Ph.D

Academic Editor

PLOS ONE

Journal Requirements:

**Additional Editor Comments:**

**Line 190 - When using the term dual saccharides absorption test for the first time, put the abbreviation DSAT immediately - currently the reader may not know what this abbreviation is about.**

**Lines 453-458 and Editor’s remarks no.35 - this section should be removed as it is not based on scientific facts. Follow current WADA 2022 position and quote it (update ref 70) - Colostrum is not mentioned on this document. I agree that the reader must be informed about all aspects, but in the case of the ingredient Colostrum itself, it is repeating myths. WADA neither recommends nor prohibits it. But likewise, WADA does not recommend the use of any other supplement.**

**In the newly introduced changes, double spaces are present in several places.**

Reviewers' comments:

Reviewer's Responses to Questions

**Comments to the Author**

1. If the authors have adequately addressed your comments raised in a previous round of review and you feel that this manuscript is now acceptable for publication, you may indicate that here to bypass the “Comments to the Author” section, enter your conflict of interest statement in the “Confidential to Editor” section, and submit your "Accept" recommendation.

Reviewer #3: All comments have been addressed

2. Is the manuscript technically sound, and do the data support the conclusions?

Reviewer #3: Yes

3. Has the statistical analysis been performed appropriately and rigorously? 

Reviewer #3: Yes

4. Have the authors made all data underlying the findings in their manuscript fully available?

Reviewer #3: Yes

5. Is the manuscript presented in an intelligible fashion and written in standard English?

Reviewer #3: Yes

6. Review Comments to the Author

Reviewer #3: (No Response)

7. PLOS authors have the option to publish the peer review history of their article (what does this mean?). If published, this will include your full peer review and any attached files.

Reviewer #3: No

---

## [Author Response · Author response to Decision Letter 2]

17 Mar 2022

Please see our responses below to the most recent comments from the Editor 

Additional Editor Comments: 

1. Line 190 - When using the term dual saccharides absorption test for the first time, put the abbreviation DSAT immediately - currently the reader may not know what this abbreviation is about.

Thank you for highlighting this, it has been changed. 

2. Lines 453-458 and Editor’s remarks no.35 - this section should be removed as it is not based on scientific facts. Follow current WADA 2022 position and quote it (update ref 70) - Colostrum is not mentioned on this document. I agree that the reader must be informed about all aspects, but in the case of the ingredient Colostrum itself, it is repeating myths. WADA neither recommends nor prohibits it. But likewise, WADA does not recommend the use of any other supplement.

This section has been removed as requested. 

3. In the newly introduced changes, double spaces are present in several places.

Thank you, we have made all efforts to correct for any spacing issues.

---

## [Editor Report · Decision Letter 3]

21 Mar 2022

A SYSTEMATIC REVIEW: ROLE OF DIETARY SUPPLEMENTS ON MARKERS OF EXERCISE-ASSOCIATED GUT DAMAGE AND PERMEABILITY

PONE-D-21-15349R3

Dear Dr.Chantler,

We’re pleased to inform you that your manuscript has been judged scientifically suitable for publication and will be formally accepted for publication once it meets all outstanding technical requirements.

Kind regards,

Krzysztof Durkalec-Michalski, Ph.D

Academic Editor

PLOS ONE

---

## [Editor Report · Acceptance letter]

31 Mar 2022

PONE-D-21-15349R3 

A SYSTEMATIC REVIEW: ROLE OF DIETARY SUPPLEMENTS ON MARKERS OF EXERCISE-ASSOCIATED GUT DAMAGE AND PERMEABILITY 

Dear Dr. Chantler:

I'm pleased to inform you that your manuscript has been deemed suitable for publication in PLOS ONE. Congratulations! Your manuscript is now with our production department. 

Kind regards, 

on behalf of

Dr. Krzysztof Durkalec-Michalski 

Academic Editor

PLOS ONE